# PREDICTING TRAINING RE-EVALUATION CURVES ENABLES EFFECTIVE DATA CURRICULUMS FOR LLMS

**Shane Bergsma, Nolan Dey & Joel Hestness**
Cerebras Systems
{shane.bergsma,nolan,joel}@cerebras.net

## ABSTRACT

Data curriculums have become central to successful LLM training, yet principles governing optimal data placement remain unclear. We introduce the *training re-evaluation curve (TREC)*, a diagnostic that retrospectively evaluates training batches *using the final model weights*. The TREC characterizes how well a trained model retains training data as a function of *when* the data was encountered during training. Analyzing TRECs for models from 111M to 3.9B parameters, we show that placing high-quality data at low points on the TREC significantly improves performance. Importantly, while a TREC is initially observable only after training, we demonstrate it can be *predicted in advance* from AdamW's implicit EMA coefficients, enabling proactive curriculum design. By predicting TRECs for published training recipes, we explain prior ablations and reveal suboptimal data placements. We also align high-quality data with TREC minima in order to improve continual pre-training of a 3.9B-parameter LLM trained on 900B tokens.

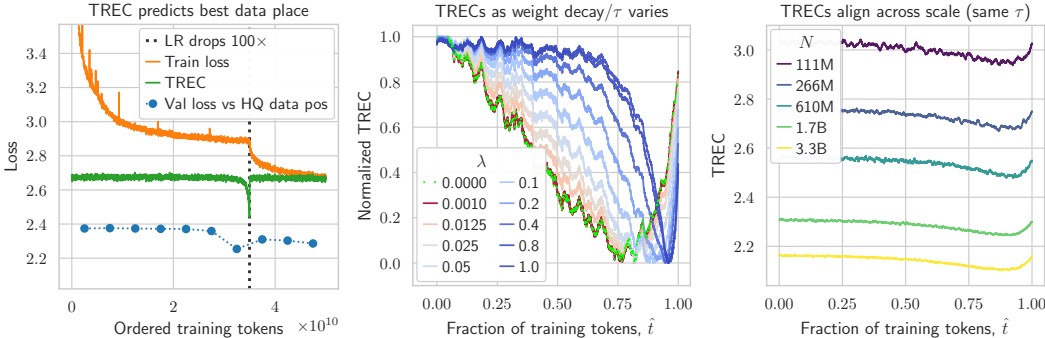

Figure 1: *Left*: (610M parameters): train loss steadily falls *after* learning rate drops to $0.01\eta_{\max}$ (after 70% of steps), but the optimal position for placing high-quality (HQ) data is *before* the LR drop, in the TREC valley. *Middle*: (610M, linear LR decay): TREC shape varies with AdamW timescale $\tau$ (varied via weight decay $\lambda$). *Right*: (size varies, linear LR decay-to-zero, all 20 tokens-per-parameter): TRECs align across 1000× scaling of training compute, when $\tau$ matches.

## 1 INTRODUCTION

LLM training now often includes *mid-training* or *annealing* phases where special data is upsampled in the final stages of pre-training. These data can be *high-quality* (Shen et al., 2024), *recent* (Dubey et al., 2024) or *domain-specific*—e.g., math (OLMo et al., 2024), code (Zhang et al., 2024a) or instructions (Hu et al., 2024). It is assumed that placing such data *at the end of training*, when learning rate is near zero, maximizes its effectiveness. While some work explores when this phase should begin (Feng et al., 2024; Parmar et al., 2024; Liu et al., 2025), "many interesting questions remain around finding the optimal dataset distribution for pre-training" (Anil et al., 2023).

We introduce the *training re-evaluation curve (TREC)*, a diagnostic for understanding LLM training and data placement. A TREC measures how well a fully trained model performs on training batches

as a function of when those batches were seen. Defined over homogeneous data, we construct it by evaluating the final model on the i.i.d. training sequence, in order. If data were retained equally, the TREC would be flat; in practice, models perform better on tokens from specific points in training.

Under *Step*-decay learning rate (LR) schedules, e.g., as used (with multi-step-decay) in DeepSeek LLM (Bi et al., 2024), the TREC may bottom-out well before the end of training (Fig. 1, *left*), while the standard training curve (on unseen batches) continues to decline. We hypothesize that high-quality (HQ) data should be placed where the TREC is lowest—that is, where the final model would achieve its lowest retrospective loss. Indeed, when retraining different models, each with the same HQ data inserted into a different 10% segment of the training trajectory, placing HQ data where the TREC is lowest yields the best validation loss (*loss-by-segment* plotted as blue points in Fig. 1, *left*).

Of course, it is impractical to train a very large model, measure its TREC, and then *re*-train with a TREC-informed curriculum. Fortunately, we show the TREC is *predictable*, enabling *proactive* curriculum design. For AdamW—the dominant optimizer in LLM pre-training—TRECs are governed by the EMA timescale (Sec. 3): AdamW parameters can be viewed as an EMA over weight updates (over data), so timescale $\tau$ controls data influence across training, and thus where TREC performance peaks (Fig. 1, *middle*). Sweeping learning rate, weight decay, or batch size with matching timescales yields identical TREC shapes (Fig. 3). Shape also persists across scale: despite $1000\times$ more training FLOPs, a 3.3B model matches a 111M model's shape (Fig. 1, *right*). In Sec. 4, we formalize a predictive model of TRECs based on an expanded view of the AdamW timescale—one that handles arbitrary LR schedules, including *Step* drops.

Based on our work, practitioners can use TRECs to guide data ordering, avoiding flawed heuristics and costly ablations. More specifically, our main contributions are:

- Introducing train re-evaluation curves, showing they **predict optimal data placement** for a given optimizer configuration (Sec. 2).
- A large-scale study of 600 TRECs in models from 111M to 3.9B parameters, and datasets from 20 to 1280 tokens-per-parameter (TPP). The study connects TRECs to the AdamW timescale (Sec. 3), and enables an analytical model for **predicting curves in advance of training** (Sec. 4).
- Explaining findings in **sparse mixture-of-experts** (Sec. 5.1) and **prior data curriculum work** (Sec. 5.2). For example, TREC prediction can explain why Llama-3 405B did not benefit, on GSM8k validation, from annealing on the GSM8k training set.
- Leveraging TRECs to improve a **3.9B-parameter LLM** trained on 900B tokens (Sec. 5.3).

## 2 TRECs PREDICT EFFECTIVE DATA PLACEMENT

We now formally define TRECs, state our key hypothesis, and describe its evaluation.

**Definition 1** (TREC). Let $\mathcal{B}_1, \ldots, \mathcal{B}_T$ denote the sequence of batches used during training, drawn independently and identically from data distribution $\mathcal{D}$, and let $\theta_T$ represent the fully-trained model parameters. The *training re-evaluation curve (TREC)* is the sequence of scalar loss values: $\mathcal{L}_{\mathrm{re}}(t) := \mathcal{L}(\mathcal{B}_t; \theta_T)$, for $t = 1, \ldots, T$, where $\mathcal{L}(\mathcal{B}_t; \theta_T)$ denotes the loss (e.g., cross-entropy) evaluated on training batch $\mathcal{B}_t$ from step $t$ using final parameters $\theta_T$. Intuitively, lower TREC loss suggests greater alignment with $\theta_T$, and may indicate $\mathcal{B}_t$ contributed more significantly to the final model.

It is worth clarifying why re-evaluation loss may depend on order, even when training batches are drawn i.i.d. In any online optimization process, each batch is encountered at a different point along the parameter trajectory, with a different amount of subsequent parameter evolution. Early batches are learned when the parameters are far from their final values, and updates derived from these batches lose their effect as the parameters subsequently evolve. The result is higher re-evaluation loss on these earlier batches: the model performs on these forgotten data similarly to how it performs on unseen validation samples. This is related to catastrophic forgetting in continual learning (McCloskey & Cohen, 1989; Kirkpatrick et al., 2017): earlier examples can be forgotten, even when the training distribution does not change. In Appendix J, we formalize this intuition and show that TREC loss can be analytically predicted as a function of parameter evolution under a simplified quadratic model.

**Definition 2** (High-Quality Data). Given training distribution $\mathcal{D}_{\mathrm{Orig}}$, a distribution $\mathcal{D}_{\mathrm{HQ}}$ is *high-quality for a task* if replacing $\mathcal{D}_{\mathrm{Orig}}$ samples with $\mathcal{D}_{\mathrm{HQ}}$ samples improves task performance.

| Schedule | Post-warmup decay $\eta(t)$ | Used in |
|---|---|---|
| Step drop at 70% of training (*Step* decay) | $\eta(t) = \begin{cases} \eta_{\max}, & t < 0.7T \\ 0.01\,\eta_{\max}, & t \geq 0.7T \end{cases}$ | Fig. 1 (*left*), Fig. 2 (*left*) |
| Linear decay to $0.1\eta_{\max}$ (10× decay) | $\eta(t) = 0.1\eta_{\max} + 0.9\eta_{\max}\left(1 - \frac{t-w}{T-w}\right)$ | Fig. 2 (*right*) |
| Linear decay to zero (D2Z decay) | $\eta(t) = \eta_{\max}\left(1 - \frac{t-w}{T-w}\right)$ | Fig. 11 (*left*) |

Table 1: Learning-rate schedules used in placement tests. All schedules share linear warmup over $w = 0.1T$ steps. Appendix Fig. 11 (right) visualizes the LR curves.

The canonical TREC is defined under homogeneous i.i.d. sampling from a base distribution $\mathcal{D}$. We average across batches when plotting TRECs in order to isolate the effect of training position from any variation in intrinsic batch difficulty. We hypothesize the resulting $\mathcal{L}_{\mathrm{re}}(t)$ primarily reflects optimization dynamics, rather than the specific $\mathcal{D}$ used to compute it. That is, given a fixed optimizer configuration, a TREC characterizes how strongly data encountered at each *training position* is retained by the final model—regardless of what data is actually inserted at those positions, or even if different data (e.g., high-quality data) is used heterogeneously across positions. Our belief in the effectiveness of this positional ranking leads to our main hypothesis:

> **Hypothesis 1**: *Placing a fixed number of high-quality samples at steps where $\mathcal{L}_{\mathrm{re}}(t)$ is lowest maximizes performance on a target task.*

**Experimental setup.** Experiments use 610M-parameter GPT2-style LLMs trained with AdamW, $\mu$P (Yang et al., 2021), and learning rate warmup over 10% of steps. After warmup, LR follows one of the decay schedules in Table 1. Full architecture and tuning details are in Appendix C. Models are trained on SlimPajama splits (Soboleva et al., 2023) (blend weights in Table 2), using the *general blend* (GB) as $\mathcal{D}_{\mathrm{Orig}}$ and *code blend* (CB) as $\mathcal{D}_{\mathrm{HQ}}$, with CB validation loss as the target task. We use 45B (90%) GB tokens and 5B (10%) CB tokens. We also construct an *aggregate blend* (AB) with a uniform 90/10 GB/CB mix. All models train on 50B tokens total (82 TPP) with no repetition. Train and TREC loss curves are computed on homogeneous AB batches.

Table 2: SlimPajama mixes used in Fig. 2 placement tests: *General Blend* is the original distribution, *Code Blend* is the HQ data.

| SlimPJ Subset | General Blend | Code Blend |
|---|---|---|
| Commcrawl | 55.1 | 37.1 |
| C4 | 28.2 | 19.0 |
| GitHub | 0.0 | 30.0[↑] |
| Books | 4.4 | 3.0 |
| ArXiv | 4.9 | 3.3 |
| Wikipedia | 4.0 | 2.7 |
| StackExch. | 3.5 | 5.0[↑] |

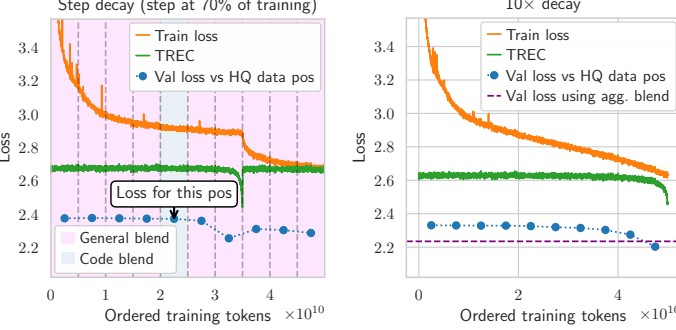

Figure 2: **TRECs predict best placement.** *Left*: Example placement curriculum 5-of-10 for the *Step* decay LR schedule. *Right*: Results for linear decay to $0.1\eta_{\max}$.

To evaluate Hypothesis 1, we conduct a data placement sweep: for each LR schedule, we train 10 models, each with the 5B CB tokens inserted into a different 10% segment of the training data. An example curriculum is shown in Fig. 2, *left*. This figure also plots the standard training curve (loss on unseen batches during training). While the *train loss* shows a familiar decrease with LR annealing (Tissue et al., 2024; Schaipp et al., 2025), note the TREC *increases* after the LR drops.

**Results.** Results in Fig. 1 (*left*, *Step* decay schedule), Fig. 2 (*right*, linear decay to $0.1\eta_{\max}$), and appendix Fig. 11 (linear decay to zero) show, in all cases, placing CB data at the step with lowest $\mathcal{L}_{\mathrm{re}}(t)$ yields the best validation loss. Data placement also outperforms uniform training on AB

(appendix Fig. 12). TRECs do not merely rediscover the *put HQ data at the end* heuristic used in prior work (with varying effectiveness, see 5.2). Looking only at the standard loss curve for *Step* decay (Fig. 1, *left*), one might think it best to "anneal" on high-quality data during the low-LR region where training loss drops rapidly. Our placement experiments show this intuition is incorrect: placing data at the TREC minimum yields better validation performance.

Under the decay-to-zero schedule (Fig. 11), the TREC minimum is likewise not at the literal end of training; the curve turns upward as the LR approaches zero. If arbitrary placement were allowed (rather than fixed 10% segments), the optimal window would occur slightly before the final steps. In practice, HQ budgets are typically far smaller than 10% of pre-training tokens, making this distinction even more consequential: sub-segments earlier within the final decile can outperform end placement. Consistent with this observation, Sec. 5.3 shows that in CPT under linear decay-to-zero, mid-CPT placement outperforms end placement across three peak learning rates.

> **Key takeaway 1**: *TRECs reliably indicate the best point for high-quality data insertion within a given optimizer configuration ($\eta(t)$, $\lambda$, B).*

Interestingly, we find that absolute $\mathcal{L}_{re}(t)$ values only weakly generalize *across different optimizer configurations* for placement prediction. Appendix D.2 investigates the predictiveness of TREC loss for placement across configurations, and discusses a grokking-vs-memorization view that may explain why configurations with similarly low TREC loss can differ in validation performance.

## 3  TREC SHAPE IS GOVERNED BY THE ADAMW TIMESCALE

To apply the insights of Sec. 2 (without having to train a model twice), we need to predict TRECs in advance. Before presenting our predictive model (Sec. 4), we build intuition for what controls TRECs (under a fixed LR schedule), finding they are mainly governed by the AdamW timescale.

**Background: the AdamW EMA and its timescale.** AdamW (Loshchilov & Hutter, 2017) updates at step $t$ can be expressed in terms of learning rate $\eta$ and weight decay $\lambda$ as: $\theta_t = (1 - \eta\lambda)\theta_{t-1} - \eta\frac{\hat{m}_t}{\sqrt{\hat{v}_t}+\epsilon}$, where $\hat{m}_t$ and $\hat{v}_t$ are bias-corrected exponentially-weighted moving averages (EMAs) of gradients and squared gradients, respectively (Kingma & Ba, 2014).

Wang & Aitchison (2024) observed that AdamW parameters $\theta_t$ can also be viewed as an EMA—of weight *updates*. Specifically, the standard EMA form $y_t = (1 - \alpha)y_{t-1} + \alpha x_t$ matches AdamW when $y_t = \theta_t$, $\alpha = \eta\lambda$, and $x_t = -\frac{1}{\lambda}\frac{\hat{m}_t}{\sqrt{\hat{v}_t}+\epsilon}$. The *timescale* $1/\alpha = 1/\eta\lambda$, denoted $\tau_{iter}$ by Wang & Aitchison, represents the approximate number of iterations over which updates are averaged.

When expressed in epochs as $\tau_{epoch} = \tau_{iter}/M$, where $M$ is the number of iterations per epoch, Wang & Aitchison found the optimal $\tau_{epoch}$ remained stable under model and dataset scaling in image tasks. Maintaining a constant $\tau_{epoch}$ requires decreasing $\lambda$ proportionally when $M$ increases.

Since LLM pre-training typically uses a single epoch, we follow Dey et al. (2025) and Bergsma et al. (2025a) in defining a normalized timescale $\tau = \tau_{iter}/T$, where $T$ is the total number of optimization steps. As $T = D/B$ (total tokens $D$ divided by batch size $B$):

$$\tau = \frac{1}{\eta\lambda T} = \frac{B}{\eta\lambda D}. \tag{1}$$

> **Hypothesis 2**: *For a given learning-rate decay schedule (Linear, Cosine, Step, etc.), the TREC is controlled by the AdamW timescale $\tau$.*

In other words, because parameters in AdamW are implicitly weighted averages over updates (derived from training data), the EMA timescale $\tau$ governs the scope of data influence on the final model, and thus the TREC. Yet higher EMA weight alone may not lower TREC loss: earlier updates can lose influence as our position on the loss surface *shifts* (discussed more in Sec. 4).

**Training fraction.** Viewing training in terms of discrete optimizer steps becomes limiting when batch sizes and sequence lengths vary. Instead, we view optimization as a continuous stochastic process over a fixed dataset, with different batch sizes yielding different discretizations. In this perspective, we plot TRECs against *training fraction* $\hat{t} = t/T = tB/D$; this naturally facilitates comparing

curves across different batch sizes, step counts, and dataset sizes. To reduce noise in small-batch settings (and align with large-batch models), we also smooth TRECs using a moving-average filter (typically over a window of 100 steps), smoothing curves without altering the underlying trajectory.

**Results.** We follow the architecture and setup of Sec. 2, but apply: (a) a linear decay-to-zero LR schedule, (b) a context length of 2048, and (c) the original GPT2 vocabulary (50257 tokens). We train on standard SlimPajama splits with original source weightings. Plot axes indicate whether the TREC is shown in absolute loss or normalized (min-max scaled) form.

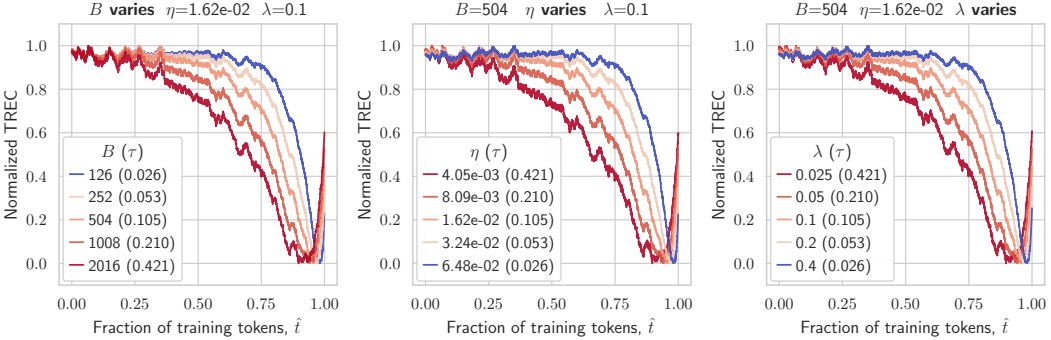

Figure 3: **Timescale $\tau$ determines TREC shape (610M, 80 TPP).** Sweeping $B$ (*left*), $\eta$ (*middle*), or $\lambda$ (*right*) produces matching variations in TRECs when $\tau$ (Eq. (1)) varies identically.

Fig. 3 shows normalized TRECs for 610M models trained to 80 TPP, sweeping $\eta$, $\lambda$, or $B$ in each subplot. Across hyperparameters, curves with matching $\tau$ exhibit very similar shapes, reflecting consistent timescale control. This alignment is especially clear here since all models use the same ordered training data. Similar patterns hold across other scales and dataset sizes. Generally, as the timescale expands ($\tau$ increases), the TREC minimum (*valley*) shifts earlier.[1]

We next examine TREC alignment as models and datasets scale. When models share the same $\tau$ and TPP, their TRECs largely align (Fig. 4, *left*), with small $\tau$ differences causing corresponding shifts. The size of *absolute* TREC drops is also similar across scales when training to the same TPP (Fig. 1, *right*). As TPP increases, the TRECs move slightly right (Fig. 4, *middle*, *right*); e.g., at 111M, increasing TPP by $64\times$ moves the TREC valley slightly later in training. Interestingly, as TPP increases, the size of the TREC drop diminishes (appendix Fig. 15): *at lower TPP, training appears to emphasize memorization*. This finding aligns with a recent study of LLM memorization from an information theoretic perspective (Morris et al., 2025).

Finally, Appendix E.3 shows that large changes in AdamW's $\beta_1$ and $\beta_2$ parameters do not significantly alter TREC shape (even when $\beta_1 = 0$, i.e., no momentum). This suggests the timescale imposed by weight decay is far more influential than the timescales of momentum and velocity.

> ***Key takeaway 2***: *Overall, the data broadly supports Hypothesis 2: the AdamW timescale ($\tau$) predominantly controls TREC shape, with TPP playing a secondary, smaller role.*

However, as seen in Fig. 2, the LR schedule itself significantly affects TREC shape. To integrate LR schedules into our analysis, we next adopt an expanded view of the AdamW timescale.

## 4 PREDICTING TRECS: ADJUSTING FOR TRAINING FRACTION

Following Sec. 3, one could in principle **predict a TREC for any model by re-evaluating a smaller model with matching $\tau$, TPP, and LR schedule**. More generally, we propose a functional form that extends the AdamW EMA perspective to time-varying learning rates, while also incorporating a training-fraction term to capture *minimizer drift*, described below.

---

[1]This shift has a limit: as $\lambda \to 0$ ($\tau \to \infty$), the curves converge (Fig. 1, *middle*). This occurs because the *shape* of the EMA *coefficients* (Sec. 4) converges to the shape of the LR schedule (Appendix F.3).

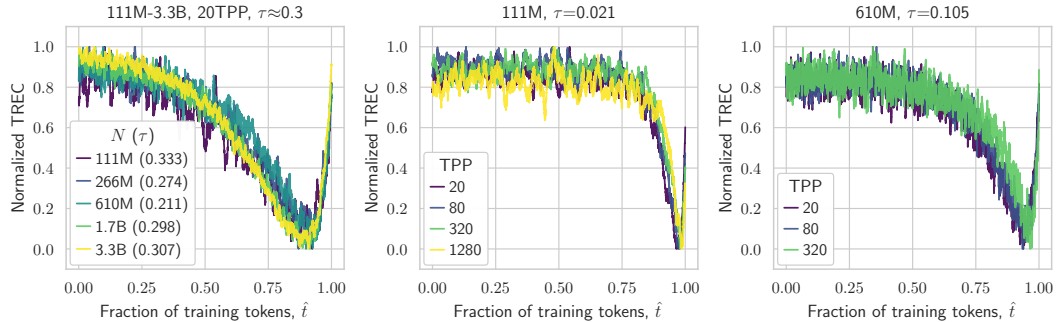

Figure 4: **Timescale $\tau$ and TREC shape across model/dataset scales.** *Left*: Similar $\tau$ yields similar TREC shapes across model scales when training to 20 TPP ($\tau \approx 0.3$). At 111M (*middle*, $\tau = 0.021$) and 610M (*right*, $\tau = 0.105$), increasing TPP shifts TRECs slightly right.

**Background: the extended AdamW EMA perspective.** Bergsma et al. (2025b) extend Wang & Aitchison (2024) by considering EMAs with time-varying smoothing parameters $\alpha_t \in [0, 1]$. Setting $\alpha_1 = 1$ (so $y_1 = x_1$), they show the recursion $y_2 = (1 - \alpha_2)\alpha_1 x_1 + \alpha_2 x_2$, and in general:

$$y_t = \sum_{i=1}^{t} \left( \prod_{j=i+1}^{t} (1 - \alpha_j) \right) \alpha_i x_i = \sum_{i=1}^{t} c_{t,i} x_i, \tag{2}$$

where $c_{t,i}$ quantifies the contribution of input $x_i$ to the output $y_t$.

With AdamW, the LR/weight decay schedule defines the time-varying smoothing $\alpha_t = \eta_t \lambda$ (Sec. 3). The EMA operates over weight updates $x_t = -\frac{1}{\lambda} \frac{\hat{m}_t}{\sqrt{\hat{v}_t} + \epsilon}$, with larger $c_{t,i}$ indicating the $i$th update contributes more to model weights $y_t = \theta_t$ at step $t$. We focus on coefficients for final model weights $\theta_T$, dropping subscript $T$ for clarity: $c_i = \eta_i \lambda \prod_{j=i+1}^{T} (1 - \eta_j \lambda)$. To connect with our continuous-time framing, we interpret $c_i$ over $T$ steps as a continuous function, reparameterizing as $c(\hat{t})$ via $\hat{t} = i/T$, enabling direct comparison with the TREC $\mathcal{L}_{\text{re}}(\hat{t})$.

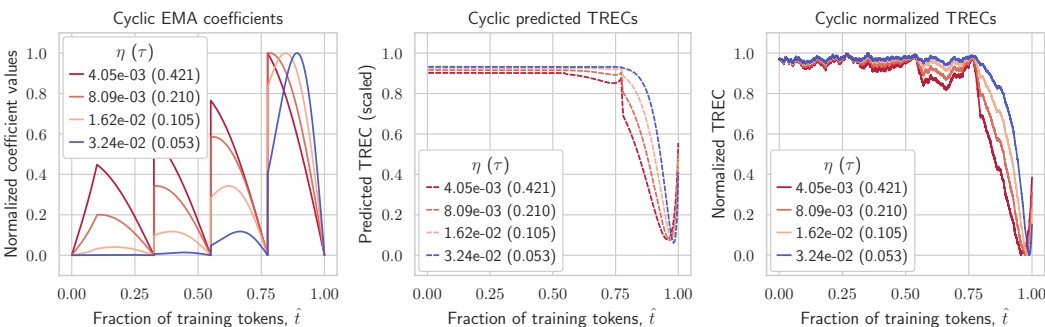

Figure 5: **Predicting TRECs (610M, 80 TPP, Cyclic LR).** *Left*: Normalized EMA coefficients $c(\hat{t})$ in our model. *Middle*: Predicted TRECs $\hat{\mathcal{L}}_{\text{re}}(\hat{t})$ from Eq. (3). *Right*: True TRECs $\mathcal{L}_{\text{re}}(\hat{t})$. Predictions match TREC dips, with early damping and late alignment with (inverted) EMA.

Fig. 5 (*left*) shows $c(\hat{t})$ for a cyclic LR schedule (schedule shown in Fig. 19). When $c(\hat{t})$ drops to zero (i.e., LR is zero), TRECs (*right*) return to baseline (1.0); when $c(\hat{t})$ is higher, TRECs dip lower. Yet $c(\hat{t})$ influence fades earlier in training, suggesting $c(\hat{t})$ alone does not fully explain TREC shape.

**Predicting TREC shape.** EMA coefficients $c(\hat{t})$ quantify each update's contribution to the final weights, but an update's *effectiveness* can fade if the batch-specific loss surface shifts after the gradient was computed, i.e., due to *minimizer drift*. Motivated by a simplified quadratic analysis (Appendix J), we model drift on a *training-fraction clock* that is scale-invariant under $\mu$P yet LR-schedule-dependent, similar to the use of *normalized compute* in Qiu et al. (2025).

> ***Hypothesis 3****: TREC shape can be predicted using the AdamW EMA coefficients combined with an adjustment for training fraction.*

Let $\mathcal{L}_{\text{re}}(\hat{t})$ denote the TREC and $c(\hat{t})$ the EMA coefficients of the final weights $\theta_T$, both indexed by training fraction $\hat{t} \in [0, 1]$. We model the normalized TREC shape using the simple form:

$$\hat{\mathcal{L}}_{\text{re}}(\hat{t}) = 1 - c(\hat{t})^p \cdot \hat{t}^m, \tag{3}$$

where $p$ and $m$ are exponents to be fit. Exponent $p$ controls the strength of the EMA contribution, while $m$ (the *training-fraction exponent*) determines when the predicted $\hat{\mathcal{L}}_{\text{re}}(\hat{t})$ begins to reflect $c(\hat{t})$. For example, with $m = 1$, fluctuations in $c(\hat{t})$ appear immediately; with a larger $m$, $\hat{\mathcal{L}}_{\text{re}}(\hat{t})$ remains near 1 for most of training and only incorporates $c(\hat{t})$ near the end.

Fig. 5 illustrates this formulation: EMA fluctuations (*left*) are dampened early in training in both predicted (*middle*, using fits of Eq. (3)) and true TRECs (*right*). We focus on predicting *shape*, not absolute values, and use Eq. (3) as a normalized functional form. We find tuning $p$ has minor impact on shape prediction compared to $m$, so we fix $p = 0.5$ across experiments and focus on fitting $m$.

**Predicting the training-fraction exponent.** For a given set of TRECs, we define the optimal $m$ as the one that maximizes shape agreement between the predicted and true TRECs. We use the Pearson correlation $r_{\text{p}}$ as a scale- and shift-invariant measure of this agreement, finding it aligns well with visual assessments (Appendix F.1). Empirically, we find that optimal exponent $m^*$ closely follows a power-law relationship with tokens-per-parameter (TPP) and the AdamW timescale $\tau$, which we express as:

$$m^* = C \cdot (\text{TPP})^{\mu_1} \cdot (\tau)^{\mu_2} \tag{4}$$

Fitting $C$, $\mu_1$, and $\mu_2$ at a small scale enables us to predict $m^*$ systematically across model/dataset sizes and hyperparameter settings, completing the components needed for TREC prediction.

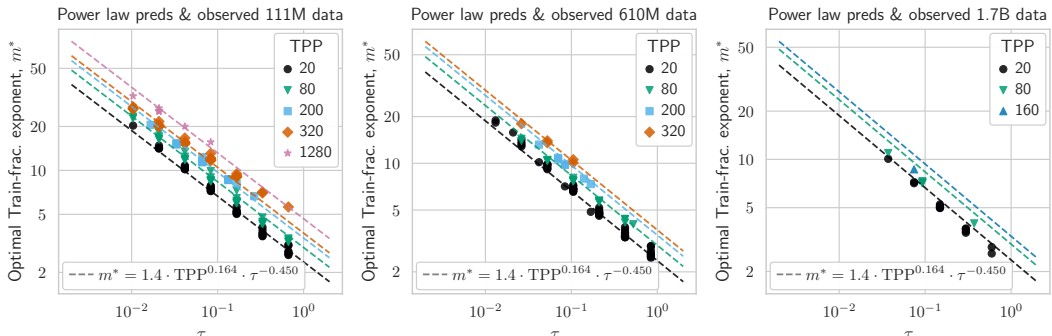

Figure 6: **Fitted power law for $m^*$ aligns with observed optima.** *Left*: 111M fit; *middle*: 610M eval; *right*: 1.7B eval. Accuracy of 111M fit holds across scale, degrading slightly for larger models.

**Results.** We follow the setup of Sec. 3, using a *Linear* decay-to-zero LR schedule unless noted. We evaluate prediction accuracy across scales and datasets using: (i) $R^2$ for predicted vs. true $m^*$ values, and (ii) Pearson $r_{\text{p}}$ for predicted vs. true TREC shape. We fit the power law Eq. (4) using $m^*$ values from small-scale 111M models, trained across varied TPPs and timescales (Fig. 6, *left*).

Fits at 111M generalize to larger scales (Fig. 6, appendix Fig. 21). As shown in appendix Table 6, while fit $R^2$ declines from 99% at 111M to 77% at 3.3B, **TREC prediction accuracy remains high across scales** ($r_{\text{p}} \sim 98\%$), confirming robust predictive performance even when $m^*$ fits are imperfect. Ablation results (appendix Table 7, fitting at 111M, evaluating at 610M scale) show that both TPP and especially $\tau$ are important for accurate $m^*$ prediction and TREC shape matching.

The observed dependence of $m^*$ on $\tau$ and TPP, and also LR schedule (Appendix G.4), matches the *minimizer drift* account in Appendix J. In the quadratic view, drift accumulates with the schedule-weighted curvature $\int_{\hat{t}}^1 \eta(s)\,h(s)\,ds$; shorter EMA timescales shorten memory and increase drift (larger $m$), higher TPP increases the extent of curvature evolution and hence drift (larger $m$), and different LR schedules directly change the cumulative integral, yielding schedule-specific $m$.

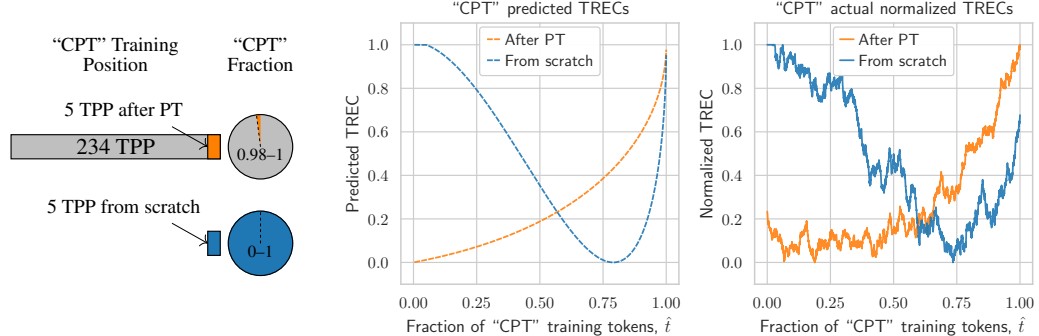

Figure 7: **Impact of training fraction on TRECs (3.9B).** *Left*: Two 3.9B models trained with identical dataset, LR, weight decay, and batch size, differing only in init (scratch vs. checkpoint after 234 TPP) and thus $\hat{t}$. *Middle*: Predicted TRECs using our framework. *Right*: True TRECs closely match predictions, showing training fraction determines shape under controlled conditions.

Fig. 7 illustrates the importance of the training fraction term in Eq. (3): two 3.9B models trained with identical dataset, learning rate, weight decay, and batch size have the same $c(\hat{t})$. However, differences in initialization (from scratch vs. pre-trained checkpoint) mean training evolves over a different $\hat{t}^m$. TREC predictions that differ solely on the basis of $\hat{t}^m$ closely match actual TRECs.

> **Key takeaway 3**: *TREC shape can be accurately predicted from EMA coefficients and a training-fraction adjustment, enabling proactive curriculum design.*

# 5 APPLICATIONS

## 5.1 APPLICATION TO SPARSE MIXTURE-OF-EXPERTS (MoE)

We now apply TREC analysis to sparse MoE architectures, where only a subset of parameters activate per input (Lepikhin et al., 2020; Fedus et al., 2022). We replace each FFN block in our 111M model with a sparse MoE layer, varying the number of experts $E$ from 1 (dense) to 32. Tokens are routed to experts via hash routing (Roller et al., 2021), ensuring balanced usage. All models train with identical total tokens and datasets. Yet each expert receives only $1/E$ of tokens, reducing an expert's *effective* tokens-per-parameter. Timescale $\tau$, however, is $E$-invariant: both batch size $B$ and total tokens $D$ scale identically with $1/E$; since $\tau = B/(\eta\lambda D)$, these reductions cancel.[2]

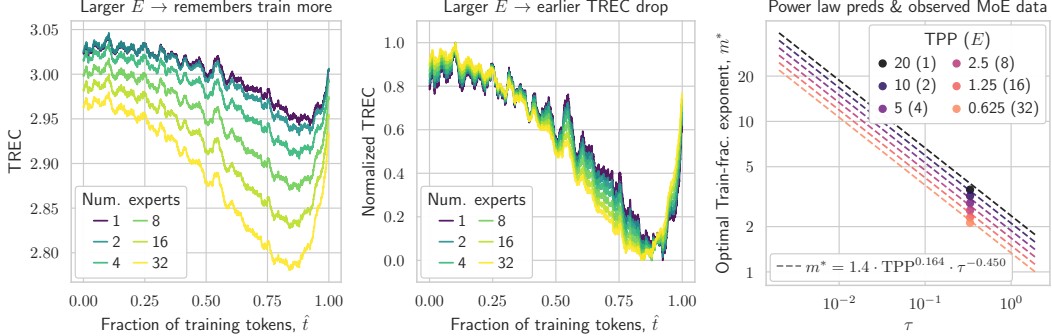

Figure 8: **Sparse MoE TRECs reflect reduced effective TPP.** *Left*: Absolute TRECs for 111M models with increasing expert count $E$ (more sparsity) and increasing TREC drop. *Middle*: Normalized curves show larger $E$ shifts the valley earlier. *Right*: $m^*$ predictions from Eq. (4) match true optima, confirming MoEs with more experts behave as if trained at reduced TPP.

---

[2]Timescale invariance may explain why correcting LR for the effectively $1/E$-smaller per-expert *batches* is unnecessary (Wei et al., 2024, Appendix C.1); MoEs might simply have won the *parameterization lottery*.

Fig. 8 shows larger $E$ produces greater and earlier TREC drops, indicating stronger memorization: MoE layers behave as if trained at their *effective* TPP. Indeed, empirically-optimal $m^*$ values for these curves align well with predictions from our dense-data power-law model (Eq. (4)), when the power law uses the *effective* TPP ($R^2 \approx 83\%$, Fig. 8-*right*). These results complement prior observations by Jelassi et al. (2024), who found that increasing experts boosts memorization more than reasoning; our analysis suggests lower effective TPP may partly drive this effect.

## 5.2 APPLICATION TO EVALUATING LLM RECIPES

While prior LLM recipes place their high-quality data at the end of training, our TREC plots suggest this strategy is suboptimal. Still, given this common practice, we ask whether the *onset* of the HQ phase—and its reported success or failure—is consistent with the predicted TREC.

Llama 3 (Dubey et al., 2024) evaluated annealing on GSM8k and MATH training sets and reported strong gains for Llama 3 8B but none for their flagship 405B model. TRECs explain this outcome: the 405B model annealed its LR from $8e-7$ to 0 over the final 40M tokens. Since the batch size is 16M, this phase spans only ∼3 optimizer steps. With such a short window and vanishing LR, EMA coefficients are essentially zero—the model retains little from this final data.

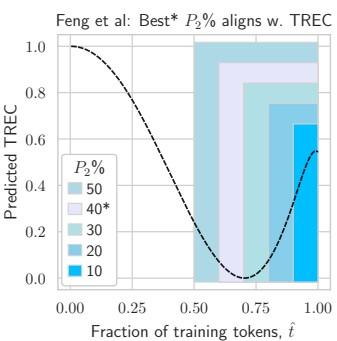 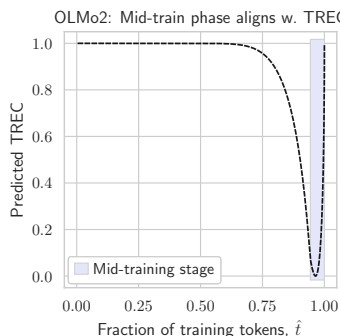 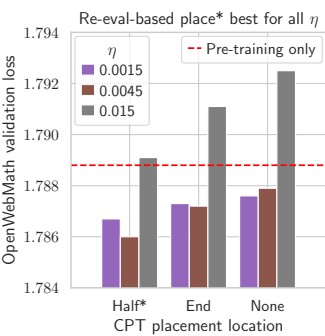

Figure 9: **Prior work agrees with predicted TRECs.** *Left*: The optimal onset point for the HQ data phase in Feng et al. (2024) aligns with the TREC bottom. *Right*: Placement of HQ data in OLMo et al. (2024) aligns with the much-narrower TREC valley.

Figure 10: **3.9B CPT results.** Placing data halfway (at TREC bottom) is most effective across all $\eta$ settings.

Fig. 9 shows two other examples. Feng et al. (2024) tested different onset points for their HQ blend, finding best results when it was used for the final 40%—a region aligning with the predicted TREC valley (*left*). In contrast, OLMo-2 13B (OLMo et al., 2024) uses its HQ blend for only the final 5.7% of training, which again aligns with the (narrower) predicted TREC dip (*right*). TREC analysis enables finding these optimal placement locations without costly trial-and-error testing.

## 5.3 CONTINUAL PRE-TRAINING OF A 3.9B LLM USING TREC INSIGHTS

We now test whether TREC-guided data placement improves outcomes in a continual pre-training (CPT) setting. Prior placement experiments in Sec. 2 inserted high-quality data partway through base-model training. In contrast, CPT typically refers to additional training performed *after* the base model is trained. To isolate this setting, we define CPT strictly as continued training *after* learning rate decay to zero—excluding works such as Parmar et al. (2024), where the CPT LR continues from a 10% decay value, thus arguably reflecting a mid-training strategy (Appendix H.3). This definition matches intuition: CPT can be performed multiple times, mid-training is only performed once.

Under this stricter definition, we take a 3.9B-parameter model trained to 234 TPP (900B tokens) with learning-rate decay-to-zero, and continue training for an additional 18B tokens (∼5 TPP; illustrated earlier in Fig. 7, *left*). During this CPT phase, we continue training on the same data blend, but insert a 1.3B-token segment of HQ data up-weighting *math*, targeting improved performance on OpenWebMath validation (Paster et al., 2023). We compare placing this HQ data halfway through CPT, where we observed the empirical TREC for the vanilla CPT run to be low (orange line in

Fig. 7, *right*), versus at the very end (as is standard), where the TREC returns to baseline. Full model, dataset, and experimental details are in Appendix I.

We test three CPT learning rates, as the optimal value is unknown. While effects are necessarily small (HQ data represents only 0.14% of total PT+CPT steps), the trend is consistent: TREC-guided placement remains effective during CPT. Placing HQ data at the TREC minimum outperforms end placement across all LRs (Fig. 10). However, when LR is too high (0.015), even correctly placed data fails to match the performance of the original base model (dashed red line on plot).[3]

## 6 RELATED WORK

Here we highlight the most relevant prior directions, with Appendix B providing full details.

**Data curriculums, influence, and attribution.** Curriculum learning explores strategies for effectively ordering training data (Bengio et al., 2009). Recent LLM curriculums often emphasize high-quality or domain-specific data in later training phases (OLMo et al., 2024; Dubey et al., 2024). While effective, crafting these approaches typically relies on heuristics or expensive experiments.

Meanwhile, quantifying influence of training points aids interpretability, auditing, and compensation (Koh & Liang, 2017; Grosse et al., 2023). Recent scalable methods, such as data value embeddings (Wang et al., 2024), recognize the importance of ordering but typically focus on retrospective attribution rather than guiding training. Memorization research examines how models retain training data, motivated by copyright or privacy concerns (Carlini et al., 2022; Schwarzschild et al., 2024).

**Re-evaluation.** Several works have analyzed loss on training data in order to probe retention dynamics. Pagliardini et al. (2024) examined memory of *specific* training batches as a function of optimizer type and LR schedule. Lesci et al. (2024) estimate memorization profiles across training and checkpoint steps for Pythia models, showing that retention depends on data order and LR, and exhibits stable cross-scale trends. Bergsma et al. (2025b) first linked retrospective losses to AdamW's EMA dynamics. In contrast, we systematically vary hyperparameters and LR schedules in order to understand, predict, and exploit data retention in LLM training.

**Scaling collapse.** Qiu et al. (2025) show that when *training loss* is normalized appropriately (and training progress is normalized similarly to $\hat{t}$), training loss curves *collapse* onto a universal trajectory across model scales. Deformations due to LR schedules are explained by a noisy-quadratic analysis. Our theoretical analysis (Appendix J) and empirical results are complementary: we study retrospective *re-evaluation* rather than initial *training* loss, account for both TPP and EMA timescale, and translate re-eval structure into actionable curriculums. In later work (Bergsma et al., 2025c), we found the same controls that govern re-evaluation loss (TPP, $\tau$, LR schedule) govern training loss curve shape.

## 7 CONCLUSION

We introduced the *training re-evaluation curve*, a simple diagnostic that evaluates how well a trained model retains individual training batches as a function of when they appeared in training. Aligning high-quality data with TREC minima improves final validation loss across models and training regimes (in both PT and CPT), surpassing the default end-of-training placement. Crucially, TRECs are largely determined by AdamW EMA coefficients and can be predicted in advance, enabling proactive data placement. We provide a theoretical account in which the training-fraction term $\hat{t}^m$ captures, in phenomenological form, the scale-invariant drift of batch-specific minimizers. In this view, the fitted $m$ is directly linked to the cumulative influence of schedule and curvature dynamics. Our insights explain existing curriculum strategies, identify suboptimal ones, and yield improved performance in large-scale continual pre-training. Taken together, our results position TREC-based placement as a principled alternative to suboptimal heuristics and costly data-onset ablations.

---

[3]We also tested inserting HQ data at the predicted TREC minimum during *pre-training* ($\sim$97%). This mid-training strategy matched the best CPT result, suggesting a rationale for why mid-training is arguably supplanting CPT in practice: it achieves comparable gains without requiring extra LR tuning or training compute.

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

## A    LIMITATIONS AND FUTURE WORK

While TRECs offer a practical and predictive diagnostic for guiding data placement in LLM training, several limitations and opportunities for future work remain.

**Optimizer scope.**    The TREC itself is not optimizer-specific: TRECs can be computed for any training run regardless of the optimization algorithm. However, our *predictive* analysis of TRECs is tailored to the AdamW optimizer, drawing on its implicit EMA formulation and corresponding timescale. Update rules for other optimizers using weight decay (such as Sophia (Liu et al., 2023, Algorithm 3) and MuonClip (Team, 2025b, Algorithm 1)) can be directly converted to an extended EMA form, exactly as was done with AdamW (Sec. 4). Moreover, the EMA perspective should also hold when AdamW is applied in a different weight basis, e.g., as in SOAP (Vyas et al., 2024), where AdamW is applied in Shampoo's eigenbasis (Gupta et al., 2018).

Extending predictive TREC models to optimizers without an implicit EMA formulation—including Adagrad (Duchi et al., 2011), Adafactor (Shazeer & Stern, 2018), or SGD variants—remains an important avenue for future exploration. For optimizers that do not use weight decay, such as Adam (Kingma & Ba, 2014), we can potentially view them as the limit of weight-decay-enhanced versions, as weight decay goes to zero. As shown in Appendix F.3, such optimizers may approach a specific EMA *shape*, even when the timescale is undefined.

**Training setups and model scales.**    Our experiments focus primarily on compute-optimal and overtrained regimes, using training runs at or beyond 20 tokens-per-parameter (TPP). While this aligns with common practices in large-scale LLM development, further work is needed to understand how TREC dynamics behave in undertrained or data-scarce settings. We also primarily study models in the 100M-4B range; exploring scaling trends for smaller or larger models could refine our conclusions.

**Data types, quality, and curriculums.**    Our work focuses on the placement of high-quality data presumed to be limited in quantity, consistent with prior observations that "truly high-quality tokens are still scarce at this moment" (Wang et al., 2025). We test placing discrete segments of HQ data, rather than continuously evolving data distributions. Moreover, although our findings generalize across blends involving code, math, and web text, we do not explicitly analyze how TRECs (or the reliability of TREC-guided placements) vary for distinct data types—such as factual vs. reasoning, or instruction vs. narrative content. A natural extension would be to combine our placement framework with recent approaches for data selection and weighting, including AutoScale (Kang et al., 2024) and RegMix (Liu et al., 2024).

Optimizing placement to maximize retention and downstream task performance raises important questions. For example, increasing memorization may come at the expense of general reasoning ability (Jelassi et al., 2024), or may confound evaluation by overly tailoring training to benchmark tasks (Dominguez-Olmedo et al., 2024). At the same time, high-quality domain-specific pre-training has been shown to be essential for models to benefit from downstream reinforcement learning (Wang et al., 2025), or SFT datasets that differ substantially from the PT distribution (Liu et al., 2025). The TREC framework provides a tool for maximizing the effect of limited training data—but whether learning from such data is ultimately *beneficial* remains an open, complex, and important question in language model research.

**Context length, vocabulary, and data diversity.**    Our study is limited to two vocabulary configurations and context lengths of 2048 and 8192 tokens. The interaction between TREC dynamics and architectural choices such as tokenizer design or sequence length remains underexplored. Similarly, we evaluate a modest range of dataset blends, and further validation is needed to assess generality across diverse languages or modalities.

**Evaluating memorization and generalization.**    TRECs measure how well the final model retains or forgets data presented at different points in training. However, we do not attempt to quantify exact memorization as defined in prior work (e.g., substring continuation (Carlini et al., 2022; Georgiev et al., 2024; Schwarzschild et al., 2024)). Future studies could examine how TREC loss relates to sequence-level memorization and whether schedules that maximize retention may inadvertently

encourage overfitting. In a similar vein, Biderman et al. (2023) offered advice on where to "place sequences that are undesirable to memorize." It is worth studying whether deliberately placing biased or undesirable data away from the TREC minimum may offer a new tool for mitigating unwanted retention. See also our notes on the "memorization window" in Appendix B.

**Predictive scope and validation.** Our predictive framework successfully anticipates TREC shape across optimizer settings and learning rate schedules. However, predictions of optimal placement for prior work are not verified via end-to-end retraining. Additionally, our CPT experiments suggest that predictive placement is most reliable within a particular optimizer configuration ($\eta(t)$, $\lambda$, $B$), and may fail to generalize across configurations. Finding effective techniques that jointly choose both placement and optimizer configuration is an open goal and an important next step. Doing so will likely require predicting when and why a model transitions from memorization into grokking, as discussed further in Appendix D.2.

Also, our predictive form focused on the (normalized) *shape* of the TREC loss (Eq. (3)), rather than its absolute *magnitude*. It would be valuable to explore other forms that can predict both shape and magnitude. It would also be valuable to expand our theoretical model to encompass these extended forms (Appendix J).

**Designing LR schedules that do not forget.** Given our predictive form for TREC loss (Eq. (3)), it is possible to design a learning rate schedule such that the EMA coefficients "cancel out" the effects of the training fraction term. That is, in theory, we may design a schedule in order to obtain a flat TREC.

In practice, such a *no-forgetting* schedule would have to rapidly decrease the LR as a power law in training steps, in order to offset the increasing power-law of the training fraction term. In prior work, such rapidly decreasing LR schedules do not perform as well as more-gradual decline (Defazio et al., 2023; Bergsma et al., 2025b). Based on the discussion in Appendix J, we may understand why: since gradients lose their effectiveness, some forgetting is actually *desirable* in LLM pre-training. Indeed, the optimal EMA timescale has been shown to decrease as a power law in TPP (Bergsma et al., 2025a), meaning that when training longer, relatively more of the data should be forgotten.

However, there are contexts where avoiding forgetting may be important (e.g., when performing CPT or SFT). It would also be interesting to investigate and mitigate sources of local optimizer drift, as a means to reducing the need for forgetting. Can we transition learning to a regime where new knowledge can be added indefinitely, without fundamentally changing the representation of such knowledge? Recent work using *lazy learning* to avoid catastrophic forgetting may provide valuable insights here for the TREC perspective (Graldi et al., 2025).

**Toward practical deployment.** Finally, while we offer actionable guidance for curriculum design (e.g., predicting TRECs in advance, avoiding late placement under step-drop or D2Z LR schedules, or leveraging homogeneous CPT phases to measure TREC dips), wider adoption will depend on usability. We will explore mechanisms to make TREC tools more performant and accessible to the community. For example, for cases where TRECs are constructed through explicit re-evaluation rather than prediction, compute can be saved by *sampling* a portion of the training batches to re-evaluate on, rather than evaluating on every batch (as we do). Determining an acceptable fraction depends on loss variance, and should be investigated systematically in future work.

## B  ADDITIONAL RELATED WORK

**Optimizing data mixtures and quality.** Recent work has explored methods for improving LLM training via better data selection or mixture strategies. One line of work focuses on identifying *what* constitutes high-quality data, including weak-to-strong selection using attention mechanisms (Hua et al., 2025), filtering based on scaling law deviations (Li et al., 2024b), or assessments from trained models (Sachdeva et al., 2024; Li et al., 2024a). Another line of work addresses *how* to mix datasets drawn from multiple domains. Ye et al. (2024) introduce *data mixing laws*, showing that validation loss can be predicted as a function of mixture proportions and proposing nested use of scaling laws to generalize to larger model/data regimes. Liu et al. (2024) similarly use small-scale training runs to regress on mixture efficacy and extrapolate to larger models, outperforming heuristic and prior-

optimized mixtures. AutoScale (Kang et al., 2024) takes a more theoretical approach, modeling how optimal domain weights vary with training scale, and deriving recipes that converge faster and perform better than baselines. While these efforts offer valuable tools for *which* data to include and *how* to weight it, our work focuses on *when* to introduce data (possibly very limited in size) during training.

**Data curriculums and annealing phases.** Our work shares conceptual motivation with curriculum learning (Bengio et al., 2009; Mindermann et al., 2022), which aims to improve generalization by presenting examples in the most effective order. In modern LLM training, this principle manifests as staged or "annealed" data mixtures, where high-quality or domain-specific datasets are introduced later in training. In Llama-3 (Dubey et al., 2024), the authors explicitly report following OpenAI's strategy (Achiam et al., 2023) of annealing on in-domain datasets like GSM8k (Cobbe et al., 2021) and MATH (Hendrycks et al., 2021).[4] Other state-of-the-art models using such practices include OLMo-2 (OLMo et al., 2024), JetMoE (Shen et al., 2024), Phi-3 (Abdin et al., 2024), Gemini (Anil et al., 2023), Gemma (Mesnard et al., 2024), MAP-Neo (Zhang et al., 2024a), Falcon-Mamba (Zuo et al., 2024) and Yi-Lightning (Wake et al., 2024). These strategies are premised on the assumption that exposing key data late in training improves downstream task performance. Recent work has also used annealing phases to assess data quality efficiently (Blakeney et al., 2024; Dubey et al., 2024; OLMo et al., 2024), enabling comparisons without full-scale pre-training. However, placing the highest-quality data at the very end is a flawed heuristic, and using ablations to determine the optimal onset for the high-quality phase is expensive. Our TREC diagnostic addresses these issues by offering a scalable method for identifying optimal data placement locations.

**Measuring loss on training data post-hoc.** As far back as Graves (2013), it was observed that generated text from a neural language model tends to inordinately reflect the final training batches; from a TREC-perspective, this manifests as lower TREC losses on the final training samples. While not the primary focus of these works, previous studies have also explicitly measured loss on previously seen training examples in order to understand model retention dynamics. In a comparison of different LR schedules, Bergsma et al. (2025b) qualitatively connected training re-evaluation losses to the exponential moving average (EMA) coefficients of AdamW updates. Pagliardini et al. (2024) propose AdEMAMix, an optimizer designed to forget more slowly. They visualize loss trajectories for individual batches across training and conclude that learning rate decay is the dominant factor controlling forgetting.

Lesci et al. (2024) introduce a difference-in-differences estimator for counterfactual memorisation, constructing a two-dimensional memorisation profile $\tau_{g,c}$ that measures the causal effect of *training on batch $g$* on *model performance at checkpoint $c$*. Their analysis shows that memorisation strength in the Pythia model suite (Biderman et al., 2023) depends on learning rate, data order, and model scale. Conceptually, our TREC can be viewed as a specific slice of such a memorisation profile: we fix $c = T$ (the final checkpoint) and measure the retrospective loss as a function of training position $g$. By not relying on public checkpoints, but instead systematically varying optimizer hyperparameters during pre-training, we develop a deeper understanding of the factors that govern retention. We also differ from all these works in that we evaluate TRECs as a tool to guide curriculum design.

**Influence estimation and data attribution.** Training data influence has long been studied for interpretability and accountability. Influence function approaches (Koh & Liang, 2017) and retraining-based approximations (Feldman & Zhang, 2020; Grosse et al., 2023) estimate data value by measuring its effect on final model behavior. Recent scalable methods such as DataInf (Kwon et al., 2023), LESS (Xia et al., 2024) and LoGRA (Choe et al., 2024) leverage gradient-based approximations (e.g., via LoRA) to approximate influence at scale. Wang et al. (2024) break from permutation-invariant assumptions and introduce trajectory-specific data value embeddings that explicitly model training data order. Their method uncovers distinct training phases in LLMs: a "high-impact warmup phase," followed by a "low-impact basin," and then a "gradual ascending" region. While insightful, this pattern differs from our findings: both EMA analysis and TREC diagnostics suggest

---

[4]While the placement details are often not reported (i.e., when exactly such data is introduced during pre-training), Dominguez-Olmedo et al. (2024) specifically identify November 2023 as a turning point, after which technical reports "start referencing certain pre-training practices that may amount to training on the test task," such as using instruction-tuning data or QA templates.

that specific batches in early training data have minimal influence on final weights. In this sense, our work serves as a valuable cross-check for attribution analyses—TRECs offer a simple, forward-only sanity-check that can validate or challenge more complex influence models.

**Memorization and forgetting dynamics.** A growing literature studies memorization in LLMs, often motivated by copyright or privacy risks. Typical methods identify memorized sequences by testing whether prompting with part of a training example elicits exact or near-exact continuations (Carlini et al., 2022; Georgiev et al., 2024; Schwarzschild et al., 2024). Morris et al. (2025) propose an information-theoretic definition of memorization based on Kolmogorov complexity, and show how it varies across dataset size, with conclusions that align with our own TREC results (Sec. 3).

In a large-scale analysis, Biderman et al. (2023) found that training order had little impact on memorization, with memorized sequences distributed approximately as a Poisson process across training. This contrasts sharply with our results: TRECs consistently show strong order effects on loss, indicating meaningful differences in retention and learning across training time. Exact sequence reproduction may be too coarse a signal to capture the subtler, gradient-based adaptations revealed by TREC analysis.

Seeking to avoid the known deleterious effects of data repetition (Hernandez et al., 2022), yet eager to make full use of their high-quality data, Team (2025a) report successful re-use of such data, so long as one "carefully estimates" and avoids the model's "memorization window." Though no further details are offered, their approach suggests another promising application of TREC analysis: it could help identify when high-quality data can be safely repeated.

While the prior works aim to avoid memorization, a separate line of work focuses on the opposite failure mode: catastrophic forgetting. This is a central concern in continual learning (Kirkpatrick et al., 2017), where models forget previously-learned knowledge when exposed to new data. Closely related is the issue of *loss of plasticity*, in which extensive pre-training reduces a model's capacity to acquire new information (Ash & Adams, 2020; Lyle et al., 2023; Dohare et al., 2024; Kumar et al., 2024; Springer et al., 2025). Our TREC framework helps disentangle these effects. For example, training to a higher TPP reduces the *magnitude* of the TREC drop (reflecting reduced plasticity), while mainly preserving the *shape* of the curve (indicating which segments are most forgotten).

**Continual pre-training (CPT) dynamics.** Continual pre-training (CPT), or lifelong learning, involves adapting models to new data distributions beyond the initial training set. Earlier work in this area emphasized domain adaptation for classification tasks (Gururangan et al., 2020; Qin et al., 2022), while recent LLM recipes apply CPT to full model continuation. However, such practices often suffer from performance degradation, even when continuing on the original domain, due to optimization challenges (Ibrahim et al., 2024; Öncel et al., 2024). CPT strategies commonly repurpose pre-training heuristics—such as annealing phases or late-stage data swaps—without principled guidance. Our work introduces a TREC framework that directly informs when and how to incorporate new data during CPT.

**Scale-stable dynamics under $\mu$P and normalized compute.** Feature-learning parameterizations such as $\mu$P can transfer hyperparameters across scale and yield early-time consistency of dynamics across widths (Vyas et al., 2023; Kalra et al., 2023), though finite-width deviations grow on harder tasks and later epochs. Complementing this, Noci et al. (2024) provide evidence of *super-consistency* in curvature (e.g., Hessian eigenvalues) along the training trajectory, supporting training fraction as a natural coordinate (Appendix J.5). Building on these observations, Qiu et al. (2025) show that when loss is indexed by *normalized compute* $x = t/t^*(p)$, where $t$ is the current step and $t^*(p)$ is the compute-optimal step count for model size $p$, then training-loss curves collapse across scales, with LR-schedule-dependent deformations explained by a noisy-quadratic analysis; collapse also holds for fixed multiples of $t^*(p)$. Their experiments, however, are limited to proof-of-concept models at relatively small scale, whereas we validate TREC dynamics in LLMs up to 3.9B parameters. Our setting also differs in target (re-evaluation vs. training loss) and mechanism (AdamW EMA timescale + drift).

In later work (Bergsma et al., 2025c), we found the same controls that govern re-evaluation loss (TPP, $\tau$, LR schedule) govern training loss curve shape.

Table 3: Model architectures used in main experiments

| Model | $d_{\text{model}}$ | $n_{\text{layers}}$ | $d_{\text{ffn}}$ | $d_{\text{head}}$ | Experiments |
|-------|--------|--------|------|-------|-------------|
| 111M | 768 | 10 | 2048 | 64 | Sec. 3 and Sec. 5.1 |
| 266M | 768 | 32 | 2048 | 64 | Sec. 3 |
| 610M | 2048 | 10 | 5461 | 64 | Secs. 2 and 3 |
| 1.7B | 2048 | 32 | 5461 | 64 | Sec. 3 |
| 3.3B | 2048 | 64 | 5461 | 64 | Sec. 3 |
| 3.9B | 2048 | 40 | 16384 | 128 | Sec. 4 (Fig. 7), Sec. 5.3 |

Table 4: Models, tokens-per-parameter (TPP) and corresponding dataset sizes (in tokens), number of model variants trained (LR schedule type, $\eta$, $\lambda$, $B$, or data placement strategy), and purpose of trained models. In total, 41 models were trained with different data placements, and 578 TRECs were computed over different optimizer hyperparameters.

| Model | TPP | $D$ | Variants trained | Purpose |
|-------|-----|-----|------------------|---------|
| 111M | 20 | 2.19B | 61 | Fitting/evaluating TREC prediction |
| 111M | 80 | 8.76B | 50 | Fitting/evaluating TREC prediction |
| 111M | 200 | 21.9B | 21 | Fitting/evaluating TREC prediction |
| 111M | 320 | 35.0B | 40 | Fitting/evaluating TREC prediction |
| 111M | 1280 | 140.1B | 11 | Fitting/evaluating TREC prediction |
| 266M | 20 | 5.31B | 25 | Fitting/evaluating TREC prediction |
| 266M | 80 | 21.2B | 19 | Fitting/evaluating TREC prediction |
| 266M | 320 | 85.0B | 19 | Fitting/evaluating TREC prediction |
| 266M | 1280 | 339.8B | 3 | Fitting/evaluating TREC prediction |
| 610M | 20 | 12.1B | 205 | Fitting/evaluating TREC prediction |
| 610M | 80 | 48.5B | 53 | Fitting/evaluating TREC prediction |
| 610M | 82 | 50.0B | 30 | Mid-training data placement tests (code blend) |
| 610M | 200 | 121.3B | 14 | Fitting/evaluating TREC prediction |
| 610M | 320 | 194.1B | 5 | Fitting/evaluating TREC prediction |
| 1.7B | 20 | 34.3B | 31 | Fitting/evaluating TREC prediction |
| 1.7B | 80 | 137.2B | 11 | Fitting/evaluating TREC prediction |
| 1.7B | 160 | 274.3B | 1 | Fitting/evaluating TREC prediction |
| 1.7B | 320 | 548.6B | 1 | Fitting/evaluating TREC prediction |
| 3.3B | 20 | 66.5B | 2 | Fitting/evaluating TREC prediction |
| 3.3B | 23 | 76.5B | 1 | Fitting/evaluating TREC prediction |
| 3.9B | 234 | 909.2B | 2 | Mid-training data placement tests (math blend) |
| 3.9B | 239 | 923.4B | 9 | Continual PT data placement tests (math blend) |

## C  EXPERIMENTAL DETAILS

Table 3 provides details on the architecture for models used in the main experiments, while Table 4 provides, for each model scale and TPP, the dataset sizes used in training, and the number of training variations explored at that scale (varying data placement strategy, or LR schedule and hyperparameters $\eta$, $\lambda$, $B$). In total, 578 TRECs were computed for the main experiments.

Further details of the 3.9B model and settings for continual pre-training experiments are in Appendix I. In the remainder of this section, we discuss the main pre-training settings.

All trained models were GPT2-style LLMs (Radford et al., 2019) with ALiBi (Press et al., 2022) embeddings and SwiGLU (Shazeer, 2020) non-linearity. We use the AdamW optimizer. Following standard practice, we do not apply weight decay or bias to LayerNorm layers. AdamW settings are $\beta_1 = 0.9$, $\beta_2 = 0.95$, and $\epsilon = 1e-8$. We report cross-entropy loss. By default we parameterize with maximal update parameterization, $\mu$P (Yang et al., 2021), with hyperparameters set via proxy tuning, as described below.

Table 5: Tuned hyperparameters for $\mu$P proxy model

| | |
|---|---|
| $\sigma_{W,\text{base}}$ | 8.67e-02 |
| $\tilde{\eta}$ | 1.62e-02 |
| $\alpha_{\text{input}}$ | 9.17 |
| $\alpha_{\text{output}}$ | 1.095 |

For a given TPP, all models have the exact same warmup phase: a linear warmup of the learning rate from 0 to the maximum value. In all runs (aside from training of the 3.9B model), warmup was 10% of the total steps. Learning rate warmup is standard practice in LLM training (Brown et al., 2020; Rae et al., 2022; Biderman et al., 2023; Dubey et al., 2024; Kosson et al., 2024).

All models in the experiments were trained on Cerebras CS-3 systems. 610M-parameter 20 TPP models take roughly 6 hours each to train on a single CS-3.

**Proxy model hyperparameter tuning.** We now describe how we tuned the $\mu$P base hyperparameters (HPs). Our proxy model is a 39M-parameter LLM with a width $d_{\text{proxy}}$ of 256, depth of 24 layers, and a head size of 64. Tuning runs were conducted on 800M tokens with $B = 256$ sequences and a context length of 2048 tokens. Tuning was performed by randomly sampling 350 configurations of base LRs, initialization standard deviations, and embedding/output-logit scaling factors. Table 5 gives the resulting top-performing values, which we used as our tuned HPs.

It is also worth noting that the LR values shown in Fig. 3, Fig. 10 and the appendix figures are the base $\mu$P LRs *before* $\mu$P-adjustment. Calculation of $\tau$ (Sec. 3) and the EMA coefficients (Eq. (2)) requires the adjusted LR (i.e., multiplying by $d_{\text{proxy}}/d_{\text{model}}$). Also, when LR decay is used, reported LR values always refer to the peak/max LR of the LR schedule.

## C.1 PLACEMENT TESTS: EXPERIMENTAL DETAILS

For the placement tests in Sec. 2, we used the standard training and validation splits of the SlimPajama dataset (Soboleva et al., 2023), but with different weighting of subsets as given in Table 2. For these experiments, we used a context length of 8192 tokens, batch sizes of 126, base peak LR of $\eta = 1.62e{-}2$ (the MUP proxy-model-tuned LR), and weight decay of $\lambda = 0.1$. We used the Llama-3 (Dubey et al., 2024) vocabulary size of 128256.

## C.2 TREC FITTING AND PREDICTION: EXPERIMENTAL DETAILS

For the experiments and analysis in Sec. 3, Sec. 4 and Sec. 5.1, we use a context length of 2048 tokens and the GPT2 (Radford et al., 2019) vocabulary of size 50257. For these experiments, we use the default source weightings for the SlimPajama dataset.

## D FURTHER DATA PLACEMENT RESULTS

We present additional experimental results supporting our finding that optimal data placement corresponds to the lowest point on the TREC. These results cover additional learning rate schedules, summarize placement effectiveness across blends, and extend validation to alternative metrics.

## D.1 WITHIN-SCHEDULE PLACEMENT OUTCOMES

Fig. 11 provides the decay-to-zero (D2Z) data placement results, along with the LR schedules for D2Z, 10×, and *Step* drop (after 70% of training); these are the three schedules tested in the Sec. 2 experiments. For D2Z, although the TRECs bend back up to baseline at the end as expected, note the tenth data placement position still obtains the lowest average TREC loss, and consequently is the optimal data placement location.

Fig. 12 isolates the results of the placement tests. Here, the y-axis gives the code blend (CB) validation loss. In each case, the *aggregate blend*, corresponding to a uniform mix of the code and

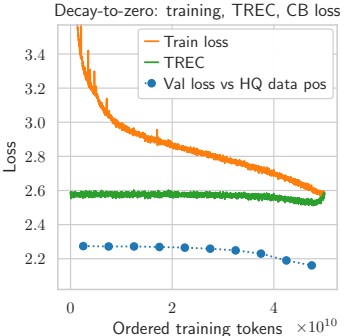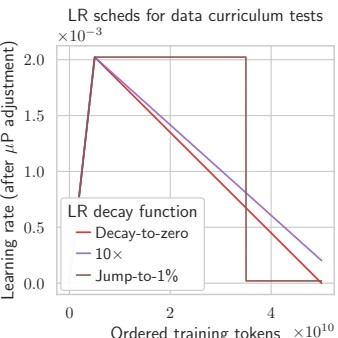

Figure 11: **TREC-guided data placement: further details.** *Left*: TRECs predict best data placement in terms of resulting CB validation loss, for a linear decay-to-zero (D2Z) LR schedule. *Right*: Plots of the D2Z, 10×, and *Step* LR schedules used in Sec. 2 (formulas in Table 1).

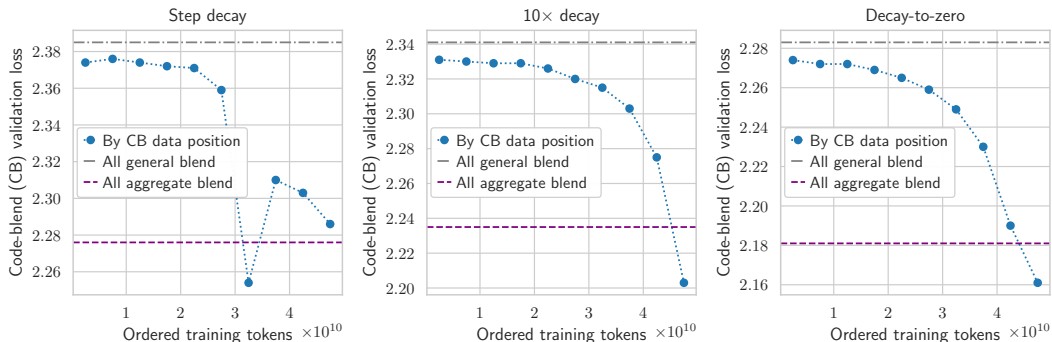

Figure 12: **Code blend (CB) validation loss across different LR schedules and CB training-data placements.** Same result data as in Fig. 1 (*left*), Fig. 2, and Fig. 11. *Left*: *Step* decay schedule, *Middle*: 10× decay, *Right*: D2Z. All 30 models that are trained with data placement improve over no placement at all ("All general blend" line), while placing in the optimal TREC-guided position always improves over the aggregate blend ("All aggregate blend" line, i.e., mixing the code blend uniformly across all training steps).

general data, is a strong baseline: it is only bested by placing the code data at the optimal placement position.

Fig. 13 presents results when all the above trained models are evaluated on the *general blend* (GB) validation data, rather than the code blend. This experiment can be interpreted as a form of ablation: if we replace GB data with *non-GB* (i.e., CB) data at a given position during training, which position leads to the greatest *degradation* in performance on GB validation (i.e., where *not* to place)? In effect, we are testing the *necessity* of GB data at each position by observing the impact of its omission. The greatest degradation consistently occurs at the position of minimum TREC loss, thus validating the placement hypothesis (Hypothesis 1) through omission rather than commission of task-relevant data.

> *Key takeaway 4*: *The TREC minimum is also the most important placement position for GB data.*

## D.2 CROSS-SCHEDULE-TREC PLACEMENT HYPOTHESIS

> *Hypothesis 4*: *Across **different** LR schedules, position-wise TREC loss predicts the effectiveness of high-quality data placement at corresponding positions.*

This hypothesis extends Hypothesis 1, proposing that TRECs are not only useful *within* a single training run but that their *absolute values* are meaningful and comparable *across* optimization sched-

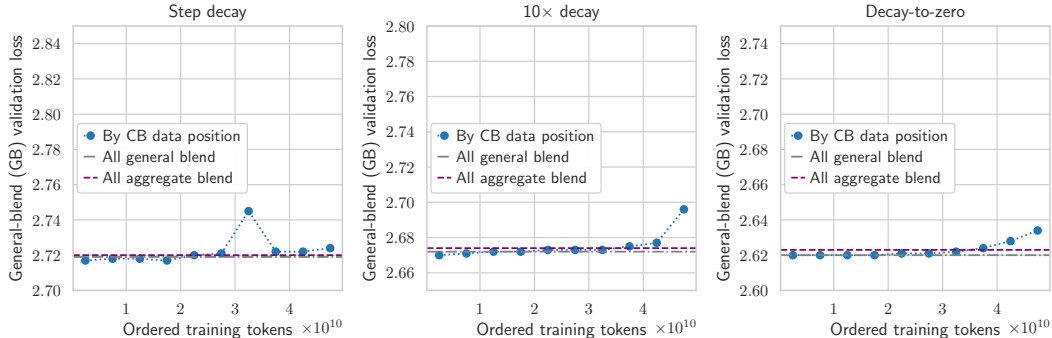

Figure 13: ***General blend* (GB) validation loss across different LR schedules and *code-blend* training-data placements.** Counterpoint to Fig. 12 (same trained models), except now the y-axis gives *loss on the **general blend***. *Left*: *Step* decay schedule, *Middle*: 10× decay, *Right*: D2Z. Placing code blend (CB) data at the TREC minimum significantly impairs performance on the general blend.

ules. For example, in the context of Sec. 2, the question is: does absolute TREC loss predict CB validation loss *across Step*, 10×, *and* decay-to-zero LR schedules?

If true, one could use predicted TRECs from multiple learning rate schedules (or weight decay/batch size configurations) to identify the globally best HQ data placement *and* optimization settings—i.e., the position, LR schedule, and other hyperparameters yielding the lowest TREC loss. Consider a 5B HQ token budget alongside 45B baseline tokens. One could scan different schedules, find the decile with the lowest predicted $\mathcal{L}_{\mathrm{re}}(t)$, and insert HQ data there. Similarly, if HQ data arrives late (e.g., recent web data to "advance the model's knowledge cut-off" (Dubey et al., 2024)), one might select the schedule with the lowest TREC loss in the *final* decile, knowing this will maximize task performance.

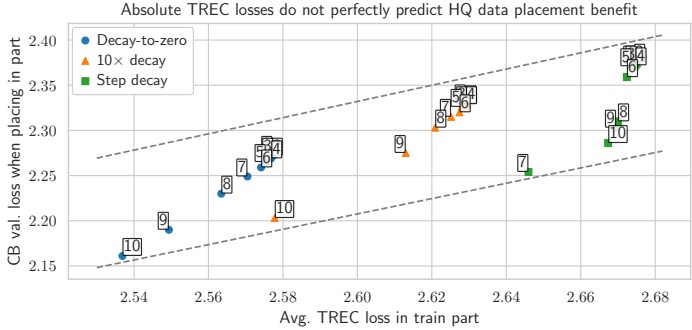

Figure 14: **Correlation between CB validation loss and TREC loss, by placement position.** CB validation loss from placement in each decile of training, versus the actual absolute TREC loss measured in that decile. Markers labeled with their deciles (1-10), dashed lines to show linear trends. While CB validation varies monotonically with TREC loss *within* one LR schedule, the correlation does not hold well *across* schedules.

Fig. 14 illustrates this idea. Each marker plots CB validation loss (y-axis) from placing HQ data in a training decile against the TREC loss (x-axis) in that decile. Markers are grouped by LR schedule, with dashed lines showing idealized linear trends.

Within each schedule, TREC loss monotonically predicts validation performance, consistent with Hypothesis 1. But across schedules, the alignment breaks down. For example, placing HQ data in the final decile of the 10× schedule yields lower validation loss than many decay-to-zero placements, despite higher TREC loss. Furthermore, large TREC differences sometimes translate to minor validation gains, and vice versa, indicating TREC loss alone does not capture all relevant dynamics.

We revisit this in Appendix I, where our 3.9B experiments show a similar disconnect: CPT segments with the lowest TREC loss (typically under high LR) do not always yield the best validation loss when HQ data is inserted. Interestingly, while TREC loss *on the placed data itself* matches TREC predictions (from the general blend), the gains do not translate to the held-out validation sets. It is worth re-iterating, however, that this generalization gap only occurs across optimizer configurations (e.g., different LRs or LR schedules), not within a given configuration.

> **Key takeaway 5**: *While TRECs reliably guide HQ placement within a given optimizer configuration, their absolute values do not consistently predict optimal placement* across *configurations. Further work is needed to guide schedule selection for data placement.*

**Toward an explanation of the generalization gap.** We can summarize the gap as follows: two schedules A and B may attain the same TREC loss at certain positions, yet A may generalize better than B. One possible explanation is that A and B can achieve similar TREC loss via qualitatively different mechanisms. Recent grokking studies (Power et al., 2022; Morris et al., 2025) show that models initially reduce training loss through memorization, and only later transition into grokking, where they learn structure that generalizes. Both mechanisms can yield equally low TREC loss, but grokking will produce stronger validation performance.

This provides a natural interpretation of why placement at position 10 of the $10\times$ schedule generalizes better than early decay-to-zero placement in Fig. 14, even though both achieve similar TREC losses: in the tenth decile of training, the $10\times$ schedule evidently leverages more grokking and less memorization. Likewise, in the CPT LR sweeps, high LR appears to push the model back into a memorization regime, while low LR preserves grokked structure and yields better validation loss.

These observations connect to classical factors known to influence generalization (such as curvature, SGD noise, and basin geometry), which may vary systematically across LR schedules. If we can identify signals that distinguish memorization-driven vs. grokking-driven TREC improvements (e.g., the width of the TREC valley, or deviations between the TREC and the standard training-loss curve), we could potentially correct for these generalization differences and enable cross-schedule prediction.

## E  FURTHER TREC RESULTS

This section provides supporting analysis for the TREC behavior discussed in Sec. 3 of the main paper. We expand on trends observed across model scale, dataset size, and tokens-per-parameter (TPP), and include additional plots that quantify absolute loss drops and timescale effects. Our findings reinforce the role of the AdamW timescale $\tau$ in shaping TRECs and offer deeper insight into how training dynamics evolve across compute regimes.

### E.1  FURTHER SCALING RESULTS

The main paper mainly focused on the *shape* and *position* of TREC valleys, as these are most pertinent for optimal data placement. As part of those findings, we found that $\tau$ and TPP both modulate the shape of the TRECs. We now examine the *absolute magnitude* of the TREC loss drops. As shown in Fig. 15, when models are trained at constant TPP (e.g., 20), the overall TREC trajectories exhibit similar total drops across scales (*left*). However, when we increase TPP while holding model scale fixed (*middle* and *right* panels), the absolute TREC drop shrinks.

This behavior is notable because it raises the hypothesis that at high TPP, models become more inertial or *rigid*—possibly due to saturation or reduced plasticity—absorbing less signal per training fraction. This aligns with recent findings on overtraining and reduced update effectiveness at high compute budgets (e.g., Kumar et al. (2024); Springer et al. (2025)). From another perspective, it could also mean that lower-TPP training focuses more on memorization, and fitting particular training examples, consistent with other recent findings (Morris et al., 2025; Jelassi et al., 2024), as discussed in Sec. 3 and Sec. 5.1.

Fig. 16 shows that even though the shape of the TREC shifts with $\tau$, the *total* (summed) TREC drop across training steps is fairly consistent (for a given TPP regime). Together with the above findings,

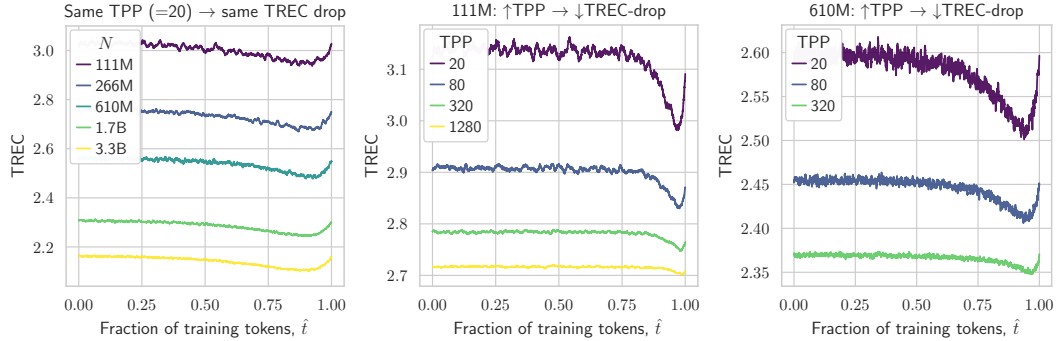

Figure 15: **Absolute magnitude of TREC drops decrease with TPP.** Plots show absolute unnormalized TRECs, same data as in Fig. 4: *Left*: $\tau \approx 0.3$; TPP is 20 for all model scales, and absolute magnitudes of drops are *similar*. *Middle*: $\tau = 0.021$, all models 111M, and magnitude of drop *decreases* with TPP. *Right*: $\tau = 0.105$, all models are 610M, and magnitude of drop again decreases with TPP.

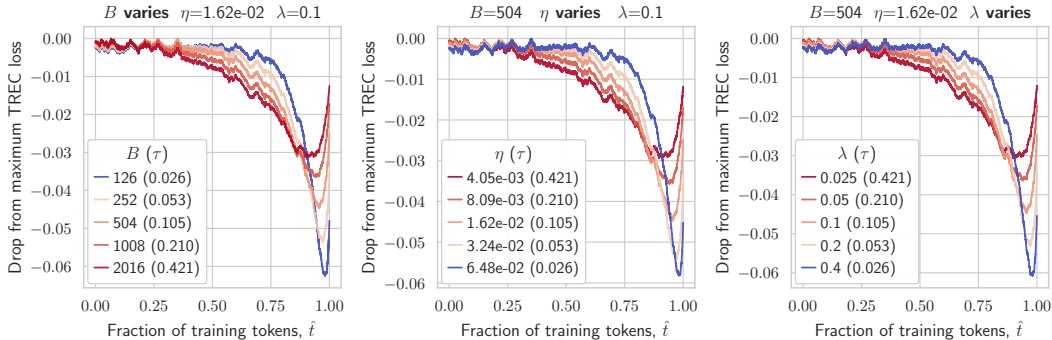

Figure 16: **Total summed TREC drop invariant to $\tau$ (610M, 80 TPP).** Same data as in Fig. 3, but showing absolute differences between the per-step and overall-maximum TREC loss. Within and across sweeps of $B$ (*left*), $\eta$ (*middle*), or $\lambda$ (*right*) the total summed TREC drop is fairly consistent. Thus the more narrow the timescale, the larger the drop.

this suggests that $\tau$ governs the *width* of the valley (how long data influences the model), while TPP sets its average *depth* (how strongly the model responds).

The upshot is that compute-efficient training regimes—like 20 TPP, where TREC valleys are sharp—may benefit most from intelligent data placement.[5] Note such regimes are often used for frontier-scale training due to their compute-efficiency. MoEs are also now often used for frontier-scale efforts, and in MoEs, expert parameters can also see relatively-low *effective* TPP (Sec. 5.1). TREC-guided data curriculums are therefore likely to offer benefits to frontier-scale training going forward.

### E.2    BATCH SIZE

We now investigate how TRECs behave across a wide range of batch sizes. In particular, we study regimes well above the *critical batch size* $B_{\mathrm{crit}}$ (McCandlish et al., 2018; Shallue et al., 2019; Merrill et al., 2025)—the point beyond which increasing batch size significantly degrades loss as a function of total tokens trained.

---

[5]Hoffmann et al. (2022) found the optimal model size $N_{\mathrm{opt}}$ and dataset size $D_{\mathrm{opt}}$ to scale roughly equally as compute increases, with the optimal $D/N$ ratio around 20 TPP. Further studies have found similar results (Besiroglu et al., 2024; Porian et al., 2024), and 20 TPP has become synonymous with compute-optimal training (Dey et al., 2023; Zhang et al., 2024b). Starting with Llama (Touvron et al., 2023), released models are often trained for more than 20 TPP because smaller, overtrained models are more efficient for *inference*.

Following Bergsma et al. (2025a), we estimate $B_{\text{crit}} \approx 2150$ tokens for 610M models trained to 20 TPP (via a fitted power law in training tokens), and we sweep batch sizes from 63 to 8064 (over two orders of magnitude). In each case, we adjust weight decay to maintain a constant AdamW timescale of $\tau = 0.421$, allowing us to isolate batch size effects from timescale variation.

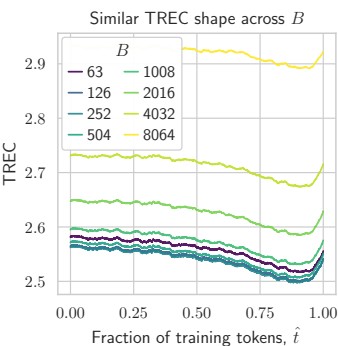 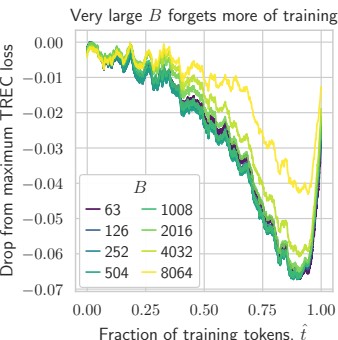

Figure 17: **TRECs and batch size** (610M, 20 TPP, $\tau = 0.421$). *Left*: Absolute TREC losses across batch sizes; very large batches achieve worse loss, but patterns are somewhat similar. *Right*: Absolute TREC losses, but normalized so they have same maximum value; $B = 8064$ behaves quite differently than all other settings.

Fig. 17 presents the resulting TRECs. The left panel shows the absolute loss curves: as batch size increases, the total TREC drop seems to become somewhat shallower, especially once batch size exceeds $B_{\text{crit}}$. The right panel aligns curves by their maximum values in order to compare shapes directly. For $B \leq 2016$, the curves remain remarkably similar in shape and position. However, for $B = 8064$ (well beyond $B_{\text{crit}}$), the curve diverges notably, indicating a qualitatively different training dynamic.

This divergence aligns with theoretical expectations: as $B$ exceeds $B_{\text{crit}}$, gradient estimates become increasingly redundant, reducing the marginal utility of each new example. From a TREC perspective, gradients from individual samples have less impact on the overall update, which focuses more on common features than idiosyncrasies of individual batches.

Viewed through the lens of TRECs, these results offer a novel diagnostic perspective on the batch size scaling frontier. These findings also suggest that models trained far beyond $B_{\text{crit}}$ may benefit less from careful data placement, as individual batches contribute less distinct signal to model updates. On the other hand, in situations where we might wish to avoid memorization (Appendix A), using larger batches could help accomplish this objective.

> **Key takeaway 6**: *TRECs remain stable across batch sizes up to $B_{\text{crit}}$, but diverge significantly beyond it, highlighting diminishing marginal data influence in large-batch regimes.*

### E.3 ADAM MOMENTUM $\beta_1$ AND VELOCITY $\beta_2$

To assess whether TREC shape is driven by other timescales in the optimizer, we sweep the AdamW momentum ($\beta_1$) and velocity ($\beta_2$) parameters while holding other hyperparameters fixed ($\tau = 0.210$, 610M model, 20 TPP). We tried a variety of settings, and report those that completed training successfully (i.e., without failure due to numerical instabilities).

Fig. 18 shows that, although absolute TREC losses vary with $\beta_1$ and $\beta_2$ (especially at extremes such as $\beta_1 = 0$ or $\beta_1 = 0.999$), TREC *shape* remains remarkably consistent across settings. This is especially evident in the right panel, where curves are normalized. Only very large momentum begins to shift the TREC minimum to slightly earlier in training.

These results reinforce the conclusion from the main text: while standard settings of $\beta_1$ and $\beta_2$ affect training dynamics and absolute loss, they do not significantly alter TREC shape. The AdamW timescale (via $\tau$) appears to be the dominant factor shaping TREC trajectories.

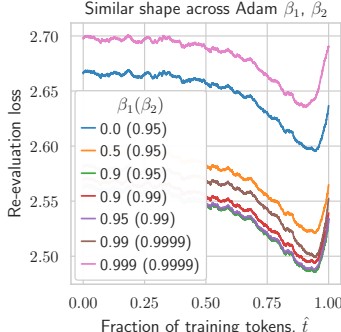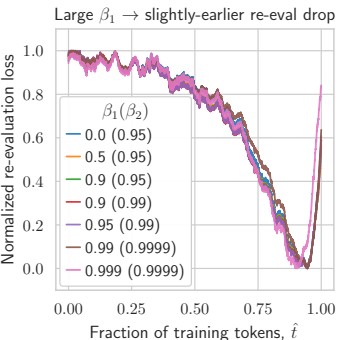

Figure 18: **TRECs and Adam $\beta_1$ and $\beta_2$** (610M, 20 TPP, $\tau = 0.210$). *Left*: Absolute TREC losses for a range of $\beta_1$ and $\beta_2$ settings. Both no momentum ($\beta_1 = 0.0$) and too much momentum ($\beta_1 = 0.999$) have highest absolute loss. *Right*: Normalized TREC losses; despite wide variation in absolute loss, normalized TREC shape is remarkably similar across all settings, with only very large momentum beginning to slightly shift the TREC minimum to earlier in training.

> ***Key takeaway 7**: TREC shape is largely invariant to changes in $\beta_1$ and $\beta_2$, underscoring the dominant role of the AdamW timescale in shaping learning dynamics.*

## F  FURTHER PREDICTION RESULTS

In this section, we provide additional details on our predictive framework for TRECs (Sec. 4), including the specific evaluation metrics used, derivations underlying key equations, and further analysis across model scales and LR schedules.

### F.1  FURTHER PREDICTION DETAILS

When computing predictions from our analytical framework (Eq. (3)), we discard the initial EMA coefficient $c_0$, as it corresponds to the influence of the model's random initialization rather than any datapoint observed during training. We focus on the remaining coefficients $c_i$ for $i \geq 1$, which quantify the contribution of training updates to the final model weights.

**Pearson correlation ($r_\mathrm{p}$).** Since we aim to predict the *shape* of TRECs $\mathcal{L}_{\mathrm{re}}(\hat{t})$ rather than absolute scale, we evaluate prediction quality using a scale- and shift-invariant metric: the Pearson correlation $r_\mathrm{p}$ between the predicted curve $\hat{\mathcal{L}}_{\mathrm{re}}(\hat{t})$ and the true curve $\mathcal{L}_{\mathrm{re}}(\hat{t})$. All plotted curves are similarly normalized to emphasize shape agreement.

Empirically, we found that Pearson correlation better aligned with human judgments of prediction quality than alternatives like $\ell_2$ loss, $R^2$, or MSE. It is computed as follows:

$$r_\mathrm{p} = \frac{\sum_t (\mathcal{L}_{\mathrm{re}}(t) - \mu_{\mathcal{L}_{\mathrm{re}}})(\hat{\mathcal{L}}_{\mathrm{re}}(t) - \mu_{\hat{\mathcal{L}}_{\mathrm{re}}})}{\sqrt{\sum_t (\mathcal{L}_{\mathrm{re}}(t) - \mu_{\mathcal{L}_{\mathrm{re}}})^2} \cdot \sqrt{\sum_t (\hat{\mathcal{L}}_{\mathrm{re}}(t) - \mu_{\hat{\mathcal{L}}_{\mathrm{re}}})^2}}, \tag{5}$$

where $\mu_{\mathcal{L}_{\mathrm{re}}}$ and $\mu_{\hat{\mathcal{L}}_{\mathrm{re}}}$ are the means of the true and predicted TREC values, respectively.

This measure ranges from $-1$ (perfect inverse correlation) to $+1$ (perfect match), with $0$ indicating no correlation.

**Illustrative example.** The middle panel of Fig. 5 illustrates the full prediction process for a *Cyclic* LR schedule. (For reference, Fig. 19, *bottom left*, shows the full LR trajectory.) We compute EMA coefficients $c(\hat{t})$ from the LR and weight decay schedule (Fig. 5, *left*), apply Eq. (3) with a fitted exponent $m$ to obtain $\hat{\mathcal{L}}_{\mathrm{re}}(\hat{t})$ (Fig. 5, *middle*), and compare it to the actual TREC (Fig. 5, *right*). The predicted curve tracks the shape of the true TREC, particularly in later training stages where the EMA contribution becomes more pronounced.

F.2 SCHEDULE HISTORY AND TRECs: *Cyclic* VS. *WSD*

In this section, we study the question: does the full LR schedule history matter, or is late-stage alignment of EMA and $\hat{t}$ sufficient to determine TREC shape?

**Motivation.** Our predictive framework assumes that TREC shape is primarily determined by the EMA coefficients and the training fraction (Eq. (3)). However, real-world schedules can differ substantially in their early phases while converging later. To test the extent to which **schedule history leaves residual effects** on TRECs, we directly compare *Cyclic* and *WSD* schedules under controlled conditions.

**Experimental setup.** We train 610M-parameter models to 80 TPP on SlimPajama using identical peak LR, batch size, weight decay, and dataset. The only difference is the LR schedule:

- The *WSD* schedule warms up, then maintains a long flat LR phase, before decaying to zero.
- The *Cyclic* schedule oscillates but aligns with *WSD* in the final 20% of training.

Because the batch size and dataset are fixed, the training fraction $\hat{t}$ at each step matches across schedules.

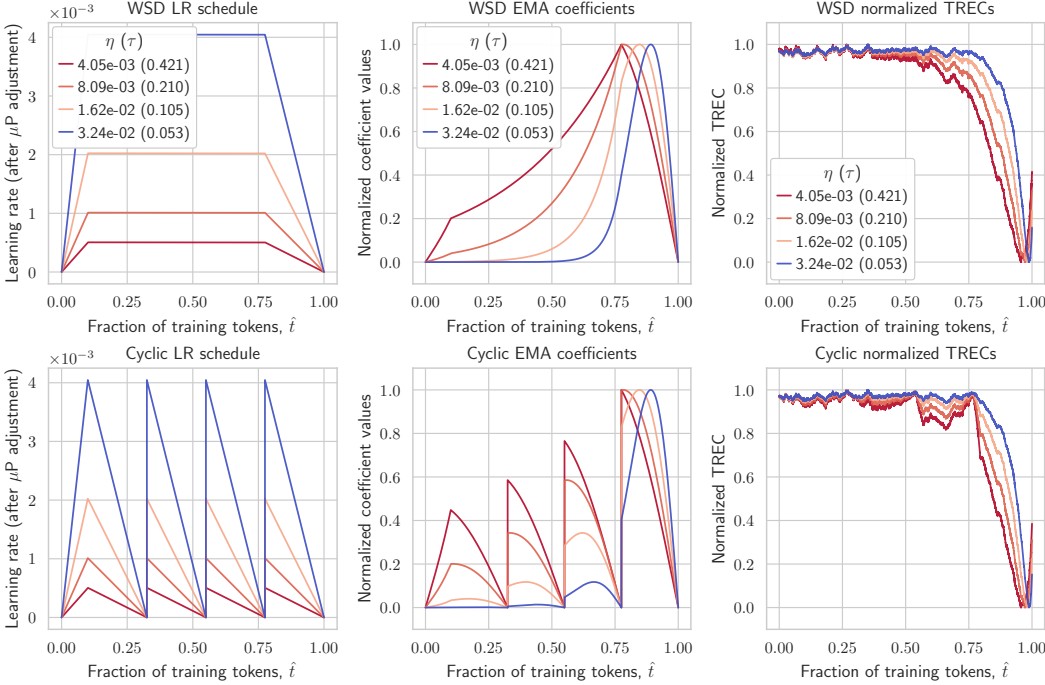

Figure 19: **Comparison of *Cyclic* and *WSD* schedules (610M, 80 TPP).** Top row: Using *WSD* schedules; bottom row: Using *Cyclic* schedules. *Left*: *LR schedule*. Training shares the same batch size, weight decay, and dataset, differing only in LR function (cyclic vs. WSD). *Middle*: *Corresponding EMA coefficients $c(\hat{t})$*. In the final 20% of training, schedules align in LR decay, resulting in identical EMA coefficients. *Right*: *TRECs*. In the final portion of training where EMA and training fraction align, TRECs also align closely, indicating that **EMA coefficients and training fraction, rather than prior LR schedule history, predominantly determine TREC shape.**

**Results.** Fig. 19 shows LR schedules (*left*), EMA coefficients (*middle*), and true TRECs (*right*) for both schedules (top: *WSD*, bottom: *Cyclic*). In the final 20% of training—where LR schedules align—the *EMA coefficients* converge exactly. Correspondingly, after a brief transient period, the *TRECs* also align, despite the differences in earlier schedule history.

In summary:

- When EMA and $\hat{t}$ align, TRECs align—even across different LR histories.
- Prior LR fluctuations leave negligible residual effect on TREC shape once EMA and $\hat{t}$ match.

> **Key takeaway 8**: *These results reinforce that given similar amounts of pre-training, **EMA coefficients and training fraction are sufficient to determine TREC shape**, supporting the generality of our predictive framework across schedules.*

### F.3 ADAM AS THE LIMIT OF ADAMW WHEN $\lambda \to 0$

We now explain why TRECs converge as $\lambda \to 0$, or equivalently, as the AdamW timescale $\tau = 1/\lambda \to \infty$. In this regime, the effect of weight decay vanishes, and AdamW behavior approaches that of standard Adam.

Fig. 1 (*middle*) illustrates this convergence at 20 TPP: the TREC for $\lambda = 0$ (vanilla Adam) closely resembles that for $\lambda = 0.001$. Since both TPP and $\tau$ are fixed, the training-fraction term $\hat{t}^m$ in Eq. (3) remains unchanged across settings, so convergence must arise from changes in the EMA coefficients $c(\hat{t})$.

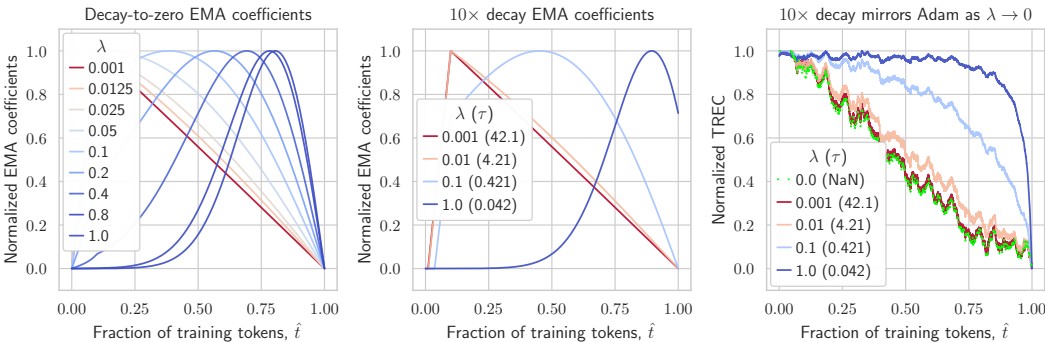

Figure 20: **Effect of weight decay on EMA coefficients and TRECs.** *Left*: Normalized EMA coefficients for the decay-to-zero LR schedule across decreasing $\lambda$ values. As $\lambda \to 0$, the EMA coefficients flatten toward the shape of the LR schedule (coefficients undefined at $\lambda = 0$). Corresponding TRECs are shown in Fig. 1, *middle*. *Middle*: EMA coefficients for the $10\times$-decay schedule. As weight decay decreases, the curve approaches the $\lambda = 0$ (Adam) case. *Right*: TRECs under the $10\times$-decay schedule. As $\lambda \to 0$, TRECs converge to the $\lambda = 0$ baseline, confirming that TREC shape becomes increasingly determined by the LR schedule alone.

Fig. 20 provides direct evidence. As shown in the *left*, decreasing $\lambda$ causes the EMA coefficients to flatten toward the shape of the LR schedule. We observe this convergence behavior both for the decay-to-zero schedule (*left*) and the $10\times$ decay schedule (*middle*), with corresponding TRECs for $10\times$ shown in the *right* panel.

In fact the EMA convergence can be derived mathematically.

**Derivation: $c_i \to \eta_i$ as $\lambda \to 0$.** To clarify the behavior of the EMA coefficients as $\lambda \to 0$ (and thus $\tau \to \infty$), it is helpful to separate the *scale* of the learning rate from its *shape*. We write the learning-rate schedule as

$$\eta_t = \gamma \, \hat{\eta}_t,$$

where $\gamma$ captures the overall scale (e.g., the peak learning rate) and $\hat{\eta}_t \in [0, 1]$ denotes the normalized schedule shape.

Under this parameterization, the EMA coefficients can be written (up to an overall scale factor) as

$$\hat{c}_i = \hat{\eta}_i \prod_{j=i+1}^{T} \left(1 - \gamma\lambda\hat{\eta}_j\right),$$

where $\hat{c}_i$ denotes the coefficients with scale factors divided out.

This form makes the limiting behavior transparent: as $\lambda \to 0$, the product term approaches 1, and therefore

$$\hat{c}_i \longrightarrow \hat{\eta}_i.$$

Thus, in the zero–weight-decay limit, the temporal structure of the coefficients is governed entirely by the learning-rate schedule shape.

An analogous limiting behavior occurs as $\gamma \to 0$ (i.e., as the overall learning-rate scale $\eta_{\max}$ vanishes), since the multiplicative term again approaches 1, yielding $\hat{c}_i \to \hat{\eta}_i$.

Moreover, the same structural behavior also emerges in the large-batch regime $B \to \infty$, with $\gamma$ and $\lambda$ held fixed. Increasing $B$ reduces the total number of optimizer steps $T$, so the product spans fewer terms and has less opportunity to compound decay. In the limit, the cumulative effect becomes negligible, and the coefficients inherit the learning-rate schedule shape.

We thank Fabian Schaipp for suggesting this scale–shape reparameterization, which clarifies these limiting regimes.

# G   Fitting the Training-Fraction Exponent $m^*$

This section provides additional details about our methodology for fitting and evaluating the power-law functional form for the training-fraction exponent $m^*$ (Eq. (4)). We describe how we select fitting data, evaluate prediction quality, validate generalization across model scales, and assess how the fit transfers across learning rate schedules.

## G.1   Data Filtering and Fit Criteria

To ensure robust and meaningful fits, we restrict our fitting dataset to regimes exhibiting well-behaved training and stable TRECs:

- **Effective $\tau$ range:** We include only runs with $\tau$ values between 0.001 and 1.0, corresponding to training runs where learning was stable and meaningful signal is present.

- **Excluding unstable hyperparameters:** We discard configurations with extremely high or low learning rates or weight decay values, which frequently result in divergence, loss spikes, or poor convergence.

- **Batch size filtering:** We remove runs where batch size exceeds the estimated critical batch size $B_{\mathrm{crit}}$, beyond which training enters a distinct large-batch regime (see Appendix E.2).

These filters ensure that the resulting fit captures relationships in the regime of effective optimization, avoiding pathological outliers.

## G.2   Evaluation Metric: $R^2$ in Log Space

We quantify the accuracy of our predicted $m^*$ values using the coefficient of determination, $R^2$, computed in log space.

Let $\{m_i\}$ denote the true optimal training-fraction exponents, i.e., those where the prediction has the highest Pearson $r_{\mathrm{p}}$ agreement with the true TRECs. Let $\{\hat{m}_i\}$ be the values predicted by our power-law fit. Then:

$$R^2 = 1 - \frac{\sum_i \left(\log m_i - \log \hat{m}_i\right)^2}{\sum_i \left(\log m_i - \overline{\log m}\right)^2} \tag{6}$$

This log-space evaluation accounts for multiplicative relationships and downweights the influence of very large or small outliers.

Table 6: **TREC** $r_{\mathrm{p}}$ **rises with scale**: even as $R^2$ with true $m^*$ declines.

| Eval scale | $m^*$: $R^2$ | TREC: $r_{\mathrm{p}}$ |
|---|---|---|
| 111M | **98.9%** | 96.6% |
| 266M | 97.2% | 97.5% |
| 610M | 98.7% | 98.4% |
| 1.7B | 89.0% | **98.7%** |
| 3.3B | 76.7% | 98.6% |

Table 7: **Both** TPP **and** $\tau$ **improve estimate of** $m^*$: fit at 111M scale, evaluation (TRECs & $m^*$ fits) at 610M.

| Fit of $m^* = C \cdot (\text{TPP})^{\mu_1} \cdot (\tau)^{\mu_2}$ | | $m^*$: $R^2$ | TREC: $r_{\mathrm{p}}$ |
|---|---|---|---|
| $\rightarrow$ uses $\tau$ | $\rightarrow$ uses TPP | | |
| ✗ | ✗ | -45.9% | 84.1% |
| ✗ | ✓ | 15.1% | 90.6% |
| ✓ | ✗ | 90.9% | 97.7% |
| ✓ | ✓ | **98.7%** | **98.4%** |

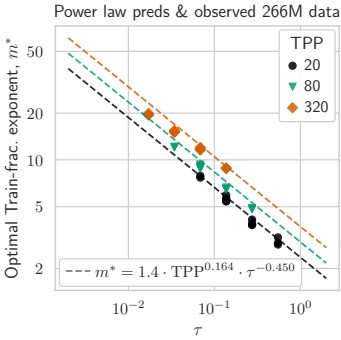
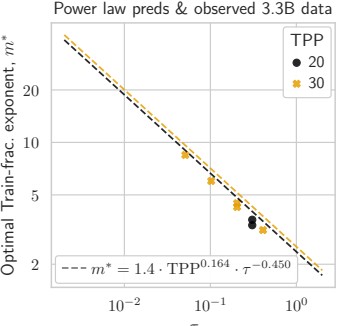

Figure 21: **Generalization of** $m^*$ **power-law fits to larger models.** We evaluate the power-law fit made at 111M scale: *Left*: Fit applied to 266M model; *right*: Fit applied to 3.3B model. While fit accuracy declines slightly at larger scale, TREC predictions remain strong overall.

### G.3 FITS AT OTHER MODEL SCALES

Fig. 21 extends the evaluation of our $m^*$ fit to additional model scales not included in the primary figure (Fig. 6). Fit quality slightly degrades at the 3.3B scale.

However, even though the optimal $m^*$ values exhibit growing prediction error (Table 6, middle column $R^2$ results), the predicted TRECs still closely match the true TREC shapes (Table 6, right column $r_{\mathrm{p}}$ results)—which is what we ultimately care about. We attribute this resilience to (1) TREC predictions being robust to exact training fraction exponents, and (2) reduced noise in the reference TRECs at larger model scales, which compensates for slightly-worse $m^*$ prediction.

Table 7 provides the ablation results mentioned in the main paper, showing how the power law performs when only a function of $\tau$, only a function of TPP, or when a function of neither—i.e., using a constant $m$ tuned over all the fitting data.

### G.4 FIT GENERALIZATION ACROSS LEARNING RATE SCHEDULES

All $m^*$ power-law fits in the main paper are derived from training runs using a *Linear* decay-to-zero LR schedule. We assess whether these coefficients generalize to other schedules by applying them to TREC predictions on models trained with a *Cosine* decay schedule.

- Using the *Linear*-based fit, we achieve a TREC prediction accuracy of $r_{\mathrm{p}} = 94.2\%$ on *Cosine*-schedule runs.
- Fitting a dedicated power-law using *Cosine* runs improves the prediction score to $r_{\mathrm{p}} = 97.8\%$.

These results indicate that while the predictive framework retains strong performance across related schedules, a schedule-specific fit yields better accuracy. Accordingly, when evaluating prior work that uses *Cosine* pre-training schedules (e.g., Sec. 5.2 and Appendix H), we employ our *Cosine*-specific $m^*$ fit in order to optimize our TREC predictions.

In summary, if a practitioner wishes to use a more complex schedule (e.g., WSD, or a custom multi-stage schedule), the *Linear*-based fit can provide strong out-of-the-box guidance. In particular,

this guidance is far stronger than the default end-of-training heuristic. For large-scale production training, hyperscalers could re-fit schedule-specific TREC predictors at small scale (analogous to how they re-fit scaling laws), and use these instead. Or, more directly, obtain a TREC prediction by simply re-evaluating a smaller model with matching $\tau$, TPP, and LR schedule.

> ***Key takeaway 9***: *Power-law prediction of $m^*$ remains robust across model scales and related schedules, though schedule-specific fits can improve accuracy. Even when $m^*$ prediction slightly degrades, the resulting TREC predictions remain highly accurate.*

## H    EVALUATING PRIOR LLM RECIPES: FURTHER DETAILS

This section provides methodological details for how we predicted TRECs for prior LLM training recipes, including Llama-3 (405B), OLMo-2 (13B), Feng et al. (8B), Pangu-Ultra (135B), and Nemotron-4 (15B).

### H.1    METHODOLOGY FOR PREDICTING TRECS OF PRIOR LLMS

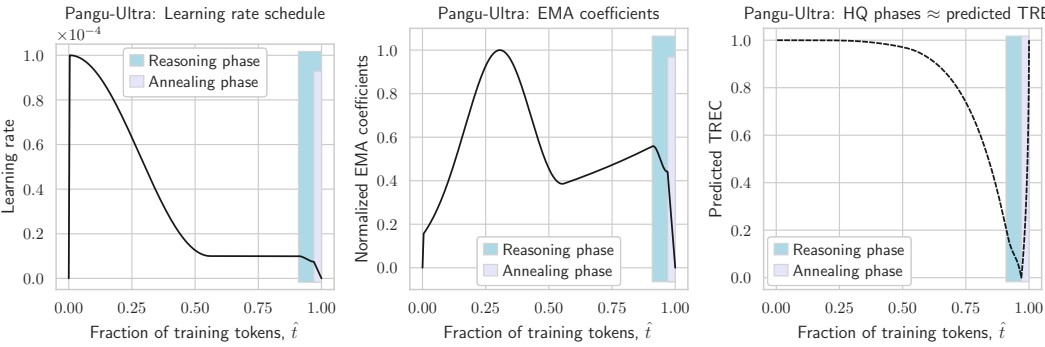

Figure 22: **Predicted TRECs for the Pangu-Ultra training recipe.** *Left*: LR schedule used in training the Pangu-Ultra model (Yin et al., 2025), including distinct "reasoning" and "annealing" phases. *Middle*: EMA coefficients computed from the LR schedule, batch size, and weight decay. *Right*: Predicted TREC. Both the reasoning and annealing phases—where high-quality data is introduced—align with the predicted TREC valley, indicating the model is well-positioned to retain this curated data.

To enable TREC prediction for existing models, we reconstructed the learning rate (LR) schedules from published training recipes. For example, Fig. 22 (*left*) shows the full LR schedule for Pangu-Ultra (Yin et al., 2025), including the designated "reasoning" and "annealing" phases. From these schedules, and given all these recipes used the AdamW optimizer, we computed the corresponding AdamW EMA coefficients (e.g., Fig. 22, *middle*).

We also extracted dataset size, batch size, sequence length, and weight decay values from the training documentation. If batch size or sequence length varied over the course of training (as is sometimes the case), we used the values applied for the majority of training. Early-stage variations typically occur well before the predicted TREC valley and thus have minimal impact on our prediction. Moreover, in continual-time interpretation, such variations do not materially affect the TREC prediction once progress is normalized to training fraction $\hat{t}$.

To determine the appropriate training-fraction exponent $m$, we applied the power-law fit described in Eq. (4), using models trained with a *Cosine* decay schedule (Appendix G.4). This was appropriate because the prior work considered here—Llama-3, Feng et al., OLMo-2, Pangu-Ultra, and Nemotron-4—all used *Cosine* LR schedules during most of pre-training. However, special mid-training phases that hold the LR constant or rapidly decay it to zero (as in Fig. 22) means that it is not exactly a *Cosine* schedule that is ultimately used over all of pre-training. Yet, while TREC prediction accuracy can benefit from a schedule-matched $m^\star$, recall that our earlier analysis has shown:

- $m^\star$ is relatively robust to schedule type,
- TREC prediction remains accurate even when $m^\star$ is slightly off, and
- our goal here is to assess general placement trends, rather than fit precise minima.

## H.2 PANGU-ULTRA RESULTS

Fig. 22 (*right*) illustrates the predicted TREC for the Pangu-Ultra training recipe (Yin et al., 2025), based on its published learning rate schedule and training phases.

The Pangu-Ultra recipe includes two late-stage phases specifically intended to improve model reasoning and instruction-following capabilities. As described in the paper:

> "In the second reasoning phase, we increase the proportion of high-quality and diverse mathematical and coding data—raising it to over 60% of the corpus to enhance the reasoning capabilities of Pangu Ultra. [...] Moreover, LLM-generated synthetic data is widely incorporated to enrich the corpus.
>
> The third annealing phase is designed to help the model consolidate and effectively apply the knowledge and reasoning skills acquired in the previous stages. Therefore, we place greater emphasis on instruction data, [...] [including] carefully refined ... short and long chain-of-thought responses."

As shown in Fig. 22 (*right*), both the reasoning and annealing phases occur during the TREC valley predicted by our framework. In some sense, this alignment between real-world data placement and our predicted optimal locations supports the validity of our framework. However, note the TRECs rise quickly during the reasoning phase, suggesting that shifting this HQ phase slightly earlier may have been beneficial. That is, a small timing adjustment could have helped avoid placing expensively-curated data in the region where the model has already begun to stabilize, minimizing the risk of diminished impact.

## H.3 ANALYSIS OF NEMOTRON-CPT (PARMAR ET AL., 2024) STRATEGY

A particularly informative paper to interpret through our framework is the recent work by Parmar et al. (2024). While their study includes a number of thoughtful ablations, it lacks a unifying theory of data placement. As such, their findings can benefit from reinterpretation in light of our insights into TRECs and training curriculums.

The starting point of their study is the Nemotron-4 15B base model, pre-trained on 8T tokens (533 TPP). Aiming to enhance this model without restarting training from scratch, they explore a continual pre-training (CPT) strategy—adding 300B further tokens, including 2.8B high-quality QA tokens mixed into a "high-quality" blend. The HQ blend is placed at the *end* of the CPT phase. Their main research questions concern: (1) the optimal onset point for introducing the HQ blend during CPT, and (2) the learning rate schedule to apply during CPT.

Although this work is framed as CPT, we first note that the original model was trained using a $10\times$ LR decay schedule, which is suboptimal relative to decay-to-zero (D2Z) (Bergsma et al., 2025b). Indeed, their own results show D2Z is a better schedule: continuing with the same pre-training data distribution but using D2Z during the 300B token extension improves downstream accuracy from 48.9% to 51.5%—a 5.3% relative gain over the base model, using less than 4% extra data. In other words, decaying to zero raises accuracy much more than would be expected by instead simply extending the $10\times$ schedule over the same number of tokens. While placing HQ data boosts performance further, the total improvement naturally reflects the *combined* effects of both D2Z and HQ data placement. Crucially, if Parmar et al. were to train further on an additional 300B tokens, they would not benefit again from applying D2Z—that benefit would already have been consumed. This is precisely why we define methods that place data *before* the LR has fully decayed to be *mid-training* rather than true CPT (Sec. 5.3): true CPT methods can be applied repeatedly (e.g., repeatedly re-warming and re-decaying the LR), while mid-training benefits, like those in Parmar et al. (2024), can only be obtained once.

Second, by placing HQ data at the very end of training, their ablations of LR schedules are intrinsically confounded. For example, they find that $100\times$ decay performs marginally better than D2Z

during the "CPT" phase. But as we have shown, D2Z TRECs rise sharply near the end, so late-stage data is unlikely to be retained. By not decaying fully, they may make better use of the (sub-optimally) placed data. Similarly, they observe that the WSD schedule (Hu et al., 2024; Hägele et al., 2024) underperforms compared to *Cosine* decay, and "hypothesize that in continued pre-training, switching the decay schedule from the one used during pre-training is harmful." However, this explanation is doubtful: different schedules yield different EMA coefficients, and thus different TRECs. Their effectiveness therefore depends not on matching schedule shapes, but on whether the HQ data is placed near the TREC minimum.

Third, their experiments are affected by data repetition, which further complicates interpretation. Repetition is known to degrade pre-training quality (Hernandez et al., 2022), and this likely applies even to high-quality data (Team, 2025a). While the subsequent work by the same team (Feng et al., 2024) controlled for repetition more carefully across training scales, the original Nemotron-CPT study in Parmar et al. (2024) allows repetition to grow significantly as the size of the CPT phase increases from 100B to 300B to 1T tokens. Specifically, the 2.8B QA tokens represent 10% of the HQ blend, implying roughly $10.7\times$ repetition if used for the full the 300B-token phase and $35.7\times$ in the 1T-token case. Their results confirm the downside of this: training with HQ data for the full CPT phase yields 53.6% accuracy at 100B tokens, but only 52.8% at 300B—i.e., longer training *hurts* when it induces excessive repetition.

For all these reasons, it is unlikely that some of the paper's specific conclusions will generalize. For instance, their recommendation that "the switch [to the high-quality] data distribution should occur at $\eta_{\max}/5$ in the learning rate schedule" is not robust across LR schedules or training regimes. More generally, our framework offers a principled alternative for determining optimal data placement that accounts for optimizer dynamics.

# I    FURTHER DETAILS ON 3.9B CPT EXPERIMENTS

## I.1    EXPERIMENTAL DETAILS

Table 3 and Table 4 provide high-level architectural and dataset details for the 3.9B model experiments. This model generally follows the experimental setup of Appendix C, with a few noteworthy exceptions detailed here. The 3.9B model is a GPT2-style LLM (Radford et al., 2019) using ALiBi (Press et al., 2022) position embeddings and a squared-ReLU (So et al., 2021) activation function. As shown in Table 3, it employs an unusually wide FFN dimension ($8\times d_{\mathrm{model}}$), which we found to be effective in early experiments.

Both pre-training and CPT phases use the AdamW optimizer with $\beta_1 = 0.9$, $\beta_2 = 0.95$, and $\epsilon = 10^{-16}$. The pre-training warmup spans 375M tokens (not 10% of total tokens), followed by linear decay to zero learning rate—also used in the CPT phase. Training is parameterized via the CompleteP variant of $\mu$P (Dey et al., 2025), with hyperparameters tuned using a depth-32 proxy model. The model uses a context length of 8192 tokens and a batch size of 672 sequences. Weight decay is set to $\lambda = 7.9 \times 10^{-4}$ based on a projection for optimal $\tau$ following Bergsma et al. (2025a). We also apply the layerwise weight decay correction from Dey et al. (2025, Table 1). The peak learning rate during pre-training is $\eta = 0.15$ (selected via proxy tuning), and we compare three different CPT learning rates (10%, 3%, and 1% of the peak PT learning rate); the specific results for each are noted in the corresponding figures.

The data blends used in these experiments are listed in Table 8. The general blend (GB) is used exclusively during pre-training and as background data in CPT. The math blend (MB), which heavily up-weights OpenWebMath, serves as our designated high-quality (HQ) data and is inserted at different positions during CPT to assess TREC-guided placement strategies, as noted in Sec. 5.3. The MB phase is 234 steps, which comprise the final 234 steps of the 3303-step CPT phase when placing at the *End*, while placement at *Half* starts at step 1980. Placement results were given previously in Fig. 10.

## I.2    EFFECT OF LEARNING RATE ON TREC LOSS DURING CPT

Fig. 23 (*left*) shows the predicted TRECs for the 3.9B model (PT+CPT), zoomed-in to show the final portion of pre-training and including the CPT phase (shown in blue shading). The depth of the

Table 8: SlimPJ and other sources used in training the 3.9B model. General Blend (GB) is used in pre-training and continual pre-training phases, while Math Blend (MB), serving as high-quality data via up-weighting of OpenWebMath, is evaluated after placing it at particular locations during CPT.

| Subset | General Blend (GB) | Math Blend (MB) |
|---|---|---|
| GitHub | 3.83% | 3.07% |
| Books | 3.62% | 2.89% |
| ArXiv | 4.21% | 3.37% |
| Wikipedia | 3.16% | 2.53% |
| StackExchange | 2.67% | 2.14% |
| Fineweb-Edu | 64.75% | 51.80% |
| Cosmopedia | 4.66% | 3.73% |
| OpenWebMath | 1.88% | 28.18%$^{\uparrow}$ |
| UltraTextBooks-2.0 | 0.42% | 2.29%$^{\uparrow}$ |
| StarCoder | 10.79% | 0.0% |

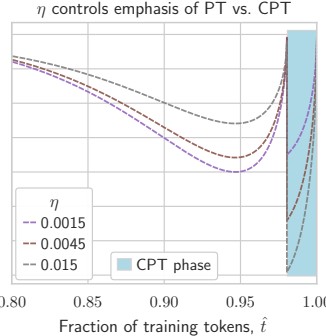 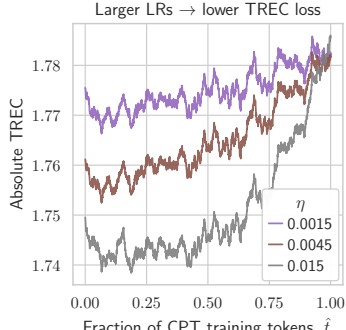

Figure 23: **3.9B CPT: higher $\eta$ in CPT leads to lower TREC loss.** *Left*: Predicted TRECs for combined PT+CPT (zoomed into final 20%). *Right*: Actual TRECs in CPT phase: $\eta = 0.015$ has largest drop.

TREC valley varies with the CPT learning rate: as $\eta$ increases, the minimum TREC loss becomes deeper. For both SFT and CPT contexts, we hypothesize that TREC predictions can help suggest the optimal post-training *learning rate* (or at least, the optimal range over which LR should be swept). We are exploring this further in ongoing work.

The observed TRECs from the actual training runs (Fig. 23, *right*) match the predicted trajectories (in blue shaded area) closely.

Interestingly, although $\eta = 0.015$ produces the deepest TREC valley, it results in the *worst* validation performance (Fig. 10). This illustrates a key failure mode for Hypothesis 4: while TREC-guided placement is robust *within* a given training schedule, it does not reliably generalize *across* learning rate schedules.

We investigate this issue further in Fig. 24, where we now also assess the TREC losses on the *placed data segments*—i.e., the HQ data that was positioned at specific CPT points (either halfway through CPT or at the end). Fig. 24 (*left*) shows that the TREC loss on the general blend (in the same segment) correlates nearly perfectly with the loss on the placed math blend, regardless of the LR or where the data was inserted. This confirms that TREC behavior generalizes from *homogeneous* to *heterogeneous* data schedules: the curve retains its predictive structure even when the inserted data differs in content.

Given this strong correlation, the critical question becomes whether performance on the *training* HQ data (math blend) generalizes to the *validation* HQ data. This is assessed in Fig. 24, *right*. Here, we observe two distinct regimes. For CPT runs using lower peak learning rates ($\eta \leq 0.0045$), there is a strong linear relationship between TREC loss and validation loss on the math blend. However, at $\eta = 0.015$, validation performance deviates from this trend entirely.

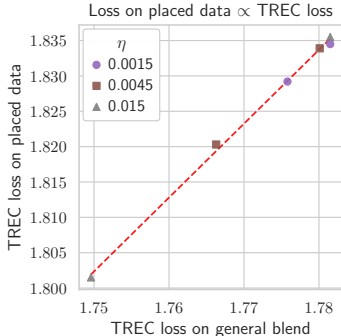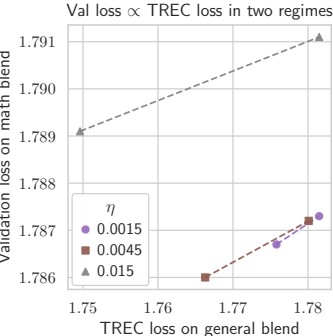

Figure 24: **3.9B CPT: Predictive power of TRECs *across* LR schedules.** *Left*: TREC loss on positions in general blend (training without placement) correlates very well with TREC loss on the *inserted/training* math blend (i.e., when this math blend data is inserted at those positions, i.e., *Half* and *End* placement). *Right*: TREC loss on positions in general blend correlates with *validation* loss on math blend data (i.e., when separate training math blend data is inserted at those positions)— when $\eta <= 0.0045$; large $\eta$ seems to induce another TREC loss regime, with different correlation with validation performance.

We speculate that high learning rates may push the model into a different region of the optimization landscape—possibly one that emphasizes memorization or shallower features—disrupting the correspondence between training and validation loss. This phenomenon is intriguing and merits further study in future work.

## J  THEORETICAL ANALYSIS OF TRECS

### J.1  MOTIVATION AND KEY CONCEPTS

Sec. 4 showed that TREC shapes are predicted well by the AdamW EMA coefficients, but only after adjusting for training fraction. Why is this adjustment needed?

Our key hypothesis is that the *effectiveness* of a gradient update depends on *where in parameter space* the update is implicitly applied. An update that was useful *when computed* may be less effective later in training if the local minimum for its batch has shifted. We argue that the pace of this minimizer drift is largely scale-invariant and governed by the training fraction.

More formally, each training batch $\mathcal{B}_t$, as an empirical sample, defines a corresponding loss surface $\ell(\mathcal{B}_t, \theta)$ over model parameters $\theta$. This surface is fixed by the batch itself (the data, the model architecture, and the parameter space do not change), but our position on the surface, $\theta_i$, changes over training. Fig. 25 illustrates: at $t = 100$, a gradient on $\mathcal{B}_t$ moves $\theta$ toward that batch's minimum. At $T = 1000$, reapplying the same update from a different position can increase loss, because the optimal direction has changed.

This perspective follows naturally from the EMA view of AdamW (and SGD): the final parameters $\theta_T$ are a weighted sum of earlier updates, effectively "applying" each update to the final model state, even if the update was computed long before. Batches nearer the end of training tend to have gradients more aligned with $\theta_T$, and hence the gradients on these batches retain more of their original usefulness. That is, when applied at step $T$, these gradients are more effective in terms of lowering loss on their original batches, which manifests as lower loss upon re-evaluation (using parameters $\theta_T$).

In the remainder of this section, we formalize this idea using a simple quadratic model that allows us to isolate the factors—EMA weighting and minimizer drift—that govern TREC loss. We proceed in three steps:

1. Begin with SGD, showing how it accumulates local minimizers via EMA-like dynamics.

2. Derive TREC loss as a function of EMA weights and minimizer drift.

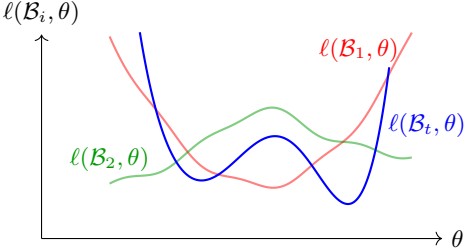

(a) Setup: Each training batch defines a distinct empirical loss surface over model parameters $\theta$. We focus on batch $t = 100$ (blue curve).

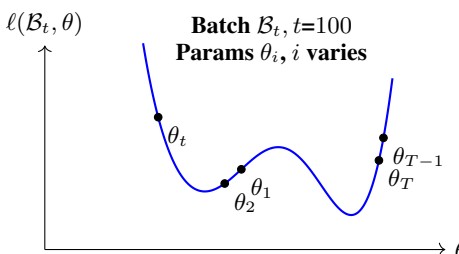

(b) Fixed loss surface, changing $\theta_i$: loss on batch $\mathcal{B}_t$ changes as parameters are updated (via gradients *on other batches*).

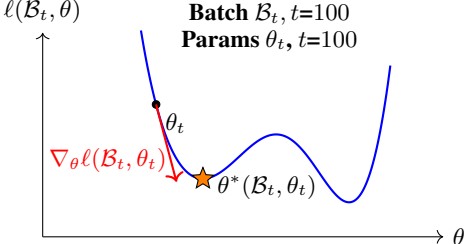

(c) Training on $\mathcal{B}_t$: at step $t$ of training, following the gradient $\nabla_\theta \ell(\mathcal{B}_t, \theta)\big|_{\theta=\theta_t}$ moves $\theta$ toward a batch-specific *and $\theta_t$-specific* local minimum.

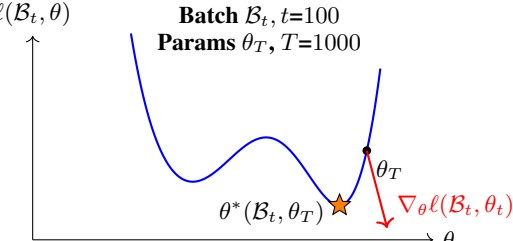

(d) Re-evaluation: applying the earlier gradient to the final model parameters $\theta_T$ now hurts batch-specific loss.

Figure 25: Illustration of how parameters and gradients affect batch-specific loss: (a) batch $\mathcal{B}_t$ defines its own unique loss surface $\ell(\mathcal{B}_t, \theta)$, with two local minima. (b) Each step of training defines a new set of parameters $\theta_i$, which determines the batch-specific loss. Now, the *effectiveness* of a gradient update to parameters $\theta$ depends on $\theta$'s current position: (c) during training, an update in the gradient direction (red arrow) is effective—it moves $\theta_t$ toward the local minimum $\theta^*(\mathcal{B}_t, \theta_t)$; (d) after training, when we re-evaluate $\mathcal{B}_t$, incorporating that same gradient update is *ineffective*—it now moves us away from the (new) local optimum, $\theta^*(\mathcal{B}_t, \theta_T)$. *Updates from batches seen later in training, when $t$ is closer to $T$, are more likely to improve loss on batch $\mathcal{B}_t$.*

3. Extend to AdamW, where the EMA is over preconditioned gradient updates.

We then explain why the training-fraction exponent in our predictive form should be scale-invariant, and conclude by discussing how this viewpoint aligns with the empirical evidence presented elsewhere in the paper.

## J.2 SETUP AND PRELIMINARIES

To gain insight into TREC loss, we adopt a simplified analytical model similar in spirit to the work of Zhang et al. (2019), who derived closed-form convergence and loss expressions for various optimizers under different batch sizes. As in their setup, we assume the optimizer dynamics are invariant to rotation and translation, allowing us to model loss as locally quadratic and separable across dimensions.

Specifically, we assume that each training batch $\mathcal{B}_t$ defines an empirical loss surface $L(\mathcal{B}_t, \boldsymbol{\theta})$ over model parameters $\theta$. We approximate this surface as *locally quadratic* at the current point in training.[6] Let $\boldsymbol{\theta}^*(\mathcal{B}_t)$ denote the local minimum. In $D$ dimensions, the loss at step $t$, given model

---

[6] While LLM training minimizes cross-entropy loss, it is common to perform a local quadratic approximation, i.e., a second-order Taylor expansion in the parameters, with the constant Hessian replaced by the instantaneous Hessian along the training trajectory (LeCun et al., 1989). Thus conclusions drawn from quadratic models often generalize to large, realistic networks (Zhang et al., 2019).

parameters $\boldsymbol{\theta}_t$, is:

$$L(\mathcal{B}_t, \boldsymbol{\theta}_t) = \frac{1}{2}(\boldsymbol{\theta}_t - \boldsymbol{\theta}^*(\mathcal{B}_t))^\top \cdot H_t \cdot (\boldsymbol{\theta}_t - \boldsymbol{\theta}^*(\mathcal{B}_t)) = \sum_{d=1}^{D} \ell^{(d)}(\mathcal{B}_t, \theta_t^{(d)}), \tag{7}$$

where $H_t = \operatorname{diag}(h_t^{(1)}, \ldots, h_t^{(D)})$ is a diagonal positive semi-definite Hessian (reflecting the batch-specific loss curvature), and

$$\ell^{(d)}(\mathcal{B}_t, \theta_t^{(d)}) = \frac{1}{2} \cdot h_t^{(d)} \cdot \left(\theta_t^{(d)} - \theta^{*,(d)}(\mathcal{B}_t)\right)^2 \tag{8}$$

is the per-dimension contribution to the loss.

**The key idea: "optimal" parameters can change.** To streamline the analysis, we now focus on a single dimension and drop the superscript notation. Unlike Zhang et al. (2019), we explicitly consider that **the locally-optimal parameter $\theta^*$ for a batch $\mathcal{B}_t$ may depend on the model state at a given training step**, i.e., what's optimal depends on where we are on the loss surface. We therefore denote this local optimum as $\theta^*(\mathcal{B}_t, \theta_s)$: the local minimizer for batch $\mathcal{B}_t$ as computed at step $s$; think of $\mathcal{B}_t$ as defining the loss surface, while $\theta_s$ defines our current position on it—and the local minimizer depends on our position. This is illustrated in Fig. 25, where the loss surface is fixed, but the local minimizer is different at step $s = 100$ (Fig. 25c) and step $s = 1000$ (Fig. 25d), as our position on the loss surface changes. Furthermore, for simplicity, we assume the batch-specific curvature $h_t$ remains fixed across steps.

At each step of training, the loss is evaluated at the *current* model parameters $\theta_t$ (our current position on the loss surface), so the *standard training loss curve* is:

$$\ell(\mathcal{B}_t, \theta_t) = \frac{1}{2} \cdot h_t \cdot (\theta_t - \theta^*(\mathcal{B}_t, \theta_t))^2. \tag{9}$$

However, for *TREC*, we compute loss using the *final* model parameters $\theta_T$ (our final position on the loss surface), i.e., using the same empirical loss surface, but positioned near a (potentially) different local minimizer. Thus the *TREC* is:

$$\mathcal{L}_{\mathrm{re}}(t) := \ell(\mathcal{B}_t, \theta_T) = \frac{1}{2} \cdot h_t \cdot (\theta_T - \theta^*(\mathcal{B}_t, \theta_T))^2. \tag{10}$$

We have endeavored to make this key concept clear because this distinction—between what's optimal at the time a batch was seen (Eq. (9)) and what's optimal when re-evaluating it later using the final model parameters (Eq. (10))—is central to understanding TREC dynamics. We will now examine how minibatch SGD accumulates these position-specific minimizers and how temporal shifts between $\theta^*(\mathcal{B}_t, \theta_t)$ and $\theta^*(\mathcal{B}_t, \theta_T)$ affect the TREC loss.

## J.3 SGD: TREC LOSS AND EMA COEFFICIENTS

We now derive the TREC loss under vanilla SGD and show how it relates to the EMA coefficients and shifts in local optima over time.

The gradient of the quadratic loss surface at step $t$ is:

$$\nabla_\theta \ell(\mathcal{B}_t, \theta_t) = h_t \cdot (\theta_t - \theta^*(\mathcal{B}_t, \theta_t)), \tag{11}$$

and the parameter update under SGD is:

$$\theta_{t+1} = \theta_t - \eta_t \nabla_\theta \ell(\mathcal{B}_t, \theta_t) = \theta_t - \eta_t h_t \cdot (\theta_t - \theta^*(\mathcal{B}_t, \theta_t)), \tag{12}$$

where $\eta_t$ is the learning rate at step $t$.

This can be rearranged as:

$$\theta_{t+1} = (1 - \eta_t h_t) \cdot \theta_t + \eta_t h_t \cdot \theta^*(\mathcal{B}_t, \theta_t), \tag{13}$$

which is equivalent to an exponential moving average (EMA):

$$y_t = (1 - \alpha_t)y_{t-1} + \alpha_t x_t, \tag{14}$$

where $y_t = \theta_t$, $\alpha_t = \eta_t h_t$, and $x_t = \theta^*(\mathcal{B}_t, \theta_t)$ (assuming the LR $\eta_t$ is chosen so that the sum is stable, i.e., $0 \leq \alpha_t \leq 1$).

Just as in the AdamW EMA case (Sec. 4), we can unroll the recursion and explicitly compute the contribution of each step to the final parameter value:

$$\theta_T = \sum_{i=1}^{T} \left( \alpha_i \prod_{j=i+1}^{T} (1 - \alpha_j) \right) \theta_i^*(i) = \sum_{i=1}^{T} c_i \theta_i^*(i), \tag{15}$$

where $c_i = \alpha_i \prod_{j=i+1}^{T} (1 - \alpha_j)$.

As noted above, we are primarily interested in the TREC loss of batch $\mathcal{B}_t$, evaluated at the final model parameters $\theta_T$, on a loss surface with minimizer $\theta^*(\mathcal{B}_t, \theta_T)$ (Eq. (10)). Substituting our EMA expression for $\theta_T$ into Eq. (10):

$$\ell(\mathcal{B}_t, \theta_T) = \frac{1}{2} \cdot h_t \cdot \left( \left[ \sum_{i=1}^{T} c_i \theta_i^*(i) \right] - \theta^*(\mathcal{B}_t, \theta_T) \right)^2. \tag{16}$$

To isolate the contribution from $i = t$:

$$\ell(\mathcal{B}_t, \theta_T) = \frac{1}{2} \cdot h_t \cdot \left( \left[ \sum_{i \neq t} c_i \theta_i^*(i) \right] + c_t \theta^*(\mathcal{B}_t, \theta_t) - \theta^*(\mathcal{B}_t, \theta_T) \right)^2. \tag{17}$$

In the special case where $\theta^*(\mathcal{B}_t, \theta_t) = \theta^*(\mathcal{B}_t, \theta_T)$ (i.e., the local minimizer has not changed):

$$\ell(\mathcal{B}_t, \theta_T) = \frac{1}{2} \cdot h_t \cdot \left( \left[ \sum_{i \neq t} c_i \theta_i^*(i) \right] + (c_t - 1) \theta^*(\mathcal{B}_t, \theta_T) \right)^2. \tag{18}$$

The loss in Eq. (18) is fully minimized when $c_t = 1$ and all other $c_i = 0$. More generally, the steps with the highest EMA coefficient will obtain the lowest loss: with a static local minimizer, the EMA coefficients fully define the TREC trajectory.

However, in practice, the local optimum for batch $t$ may drift over the course of training. We model this by assuming:

$$\theta^*(\mathcal{B}_t, \theta_t) = r_t \cdot \theta^*(\mathcal{B}_t, \theta_T), \tag{19}$$

with $r_t \in \mathbb{R}$ a scaling factor (possibly $< 0$). The TREC loss becomes:

$$\ell(\mathcal{B}_t, \theta_T) = \frac{1}{2} \cdot h_t \cdot \left( \left[ \sum_{i \neq t} c_i \theta_i^*(i) \right] + (c_t r_t - 1) \theta^*(\mathcal{B}_t, \theta_T) \right)^2. \tag{20}$$

This expression emphasizes that the TREC loss is minimized when $c_t r_t = 1$ (and the sum over $i \neq t$ vanishes), i.e., when the model fully incorporates the local minimizer for batch $t$ via a high $c_t$, *and* this minimizer is well-aligned with the one at the final step $T$. This analytical result mirrors our empirical finding (Sec. 4) that EMA coefficients alone are insufficient to predict TREC loss: early gradients may have high coefficients $c_t$, but the local minimizers of their corresponding batches can drift substantially over training (changing $r_t$), leading to poor retention.

### J.4 ADAMW: TREC LOSS AND PRECONDITIONED UPDATES

We obtain a similar result when analyzing AdamW with the same locally-quadratic loss surface. As derived in Sec. 4, AdamW parameters at the final step $T$ can be expressed as a convex combination of weight *updates*:

$$\theta_T = \sum_{i=1}^{T} c_i x_i, \tag{21}$$

where each update $x_i$ has the form:

$$x_t = -\frac{1}{\lambda} \cdot \frac{\hat{m}_t}{\sqrt{\hat{v}_t} + \epsilon}, \tag{22}$$

and each coefficient is defined as:

$$c_i = \left( \prod_{j=i+1}^{T} (1 - \eta_j \lambda) \right) \eta_i \lambda. \tag{23}$$

Substituting this decomposition into the TREC loss expression, we obtain:

$$\ell(\mathcal{B}_t, \theta_T) = \frac{1}{2} \cdot h_t \cdot \left( \left[ \sum_{i \neq t} c_i x_i \right] + c_t x_t - \theta^*(\mathcal{B}_t, \theta_T) \right)^2. \tag{24}$$

TREC loss is minimized when $c_t$ is large and the update $x_t$ points in the same direction as the final-step local minimizer $\theta^*(\mathcal{B}_t, \theta_T)$. In other words, it is primarily the *sign alignment* of $x_t$ and $\theta^*(\mathcal{B}_t, \theta_T)$ that determines whether the update helps or hurts TREC loss.

In the quadratic model (and optimization invariant to translation), we can assume without loss of generality that $\theta_t = 0$ at the time of the update. The gradient becomes:

$$\nabla_\theta \ell(\mathcal{B}_t, \theta_t) = h_t \cdot (\theta_t - \theta^*(\mathcal{B}_t, \theta_t)) = -h_t \theta^*(\mathcal{B}_t, \theta_t), \tag{25}$$

and assuming no momentum for simplicity, the update is:

$$x_t = \frac{1}{\lambda} \cdot \frac{h_t \theta^*(\mathcal{B}_t, \theta_t)}{\sqrt{v_t} + \epsilon} = \kappa_t \theta^*(\mathcal{B}_t, \theta_t), \tag{26}$$

where $\kappa_t \geq 0$ absorbs the preconditioning, curvature, and scaling terms.

To model drift in the loss surface, we again assume that the local minimizer for batch $t$ changes over time and satisfies:

$$\theta^*(\mathcal{B}_t, \theta_t) = r_t \cdot \theta^*(\mathcal{B}_t, \theta_T). \tag{27}$$

Substituting the change in local minimizer (Eq. (27)) into our simplified expression for the update (Eq. (26)), and using this update in the TREC loss equation (Eq. (24)) yields:

$$\ell(\mathcal{B}_t, \theta_T) = \frac{1}{2} \cdot h_t \cdot \left( \left[ \sum_{i \neq t} c_i x_i \right] + (c_t \kappa_t r_t - 1) \theta^*(\mathcal{B}_t, \theta_T) \right)^2. \tag{28}$$

This loss is reduced when the AdamW update contributes a value that matches the current minimizer direction, where $r_t$ controls the update direction (since $c_t \geq 0$ and $k_t \geq 0$). As before, loss is maximally reduced when $c_t = 1$ and the weight on all other updates vanishes. Loss is also maximally reduced when the product $c_t \kappa_t r_t \approx 1$. In practice, if the local minimizer has shifted substantially—especially if $r_t < 0$—then the early update may push in the wrong direction, *increasing* TREC loss despite having a high EMA coefficient.

This again echoes the central insight: an update's value is determined not only by its EMA weight $c_t$, but by its alignment with $\theta_T$'s position in the loss landscape.

## J.5 LOCAL MINIMIZER DRIFT AND THE TRAINING-FRACTION CLOCK

In the preceding analysis, the term $r_t$ captures the effect of *local minimizer drift*—the change in the batch-specific optimum $\theta^*(\mathcal{B}_t, \cdot)$ between when the gradient was computed (at step $t$) and when it is effectively applied (in the final model $\theta_T$). We now explain why the natural coordinate for this drift is the *training fraction* $\hat{t} = t/T$.

In the SGD case, the final parameters $\theta_T$ can be written as an EMA with time-varying smoothing $\alpha_t = \eta_t h_t$, where $\eta_t$ is the learning rate and $h_t$ the curvature along the current parameter direction

(Eqs. (13) and (15)). When $\alpha_t$ values are larger, more weight is put on recent minimizers (shrinking the EMA timescale), and parameters evolve faster from step to step. In particular, the pace at which the final parameters $\theta_T$ move away from earlier parameters $\theta_t$ is directly governed by the sequence $\{\eta_s h_s\}_{s=t}^{T}$. We naturally assume that larger movement in *parameters* coincides with larger movement in *local optimizers*, i.e., that the distance between $\theta^*(\mathcal{B}_t, \theta_t)$ and $\theta^*(\mathcal{B}_t, \theta_T)$ is also governed by this sequence.

In AdamW, the same qualitative picture holds: the learning rate enters directly into the EMA coefficients $c_t$ (Eq. (23)), while the curvature appears as a scaling factor in the updates (Eq. (26)). Although preconditioning partly normalizes curvature variation, the product of learning rate and effective curvature still governs the rate at which parameters update, and hence the rate at which local minimizers drift.

Recent work by Noci et al. (2024) shows that, under maximal update parameterization ($\mu$P), curvature statistics such as the largest Hessian eigenvalues evolve in a nearly *scale-independent* way, a phenomenon they term *super-consistency*. Qiu et al. (2025) build on this observation, hypothesizing, and providing empirical evidence, that a related curvature proxy depends only on training fraction (what they term "normalized compute") and is largely scale-independent under $\mu$P. That is, if we align two models of different sizes by their training fraction $\hat{t} = t/T$, their curvature trajectories $h(\hat{t})$ are nearly identical.

If both the curvature $h(\hat{t})$ and the learning rate schedule $\eta(\hat{t})$ are scale-independent functions of $\hat{t}$, then their product $\eta(\hat{t})h(\hat{t})$—which governs the pace of parameter drift in the quadratic model—is also scale-independent. Consequently, the sequence of curvature/learning-rate conditions experienced by a small model at $\hat{t} = 0.3$ is essentially the same as that experienced by a large model at $\hat{t} = 0.3$. Based on the prior work in $\mu$P and our own empirical results (discussed below, Appendix J.7), we speculate that in more complex settings, other aspects of training dynamics also evolve with training fraction. For example, in $\mu$P neural networks trained with cross-entropy loss, the extent of feature learning and the evolution of representations in earlier layers should also evolve invariant to model size. This yields a *training-fraction clock* for minimizer drift: relative training progress, not absolute tokens, is the natural coordinate for $r_t$ in re-evaluation dynamics. In this view, $r_t$ should be written as $r(\hat{t})$.

## J.6 FROM CURVATURE TO FUNCTIONAL FORM

Our quadratic-model analysis suggests that the product $\eta(\hat{t})h(\hat{t})$—learning rate times curvature—sets the magnitude of parameter updates, and hence the rate at which local minimizers move. A gradient computed at training fraction $\hat{t}$ will remain well aligned with the final model parameters $\theta_T$ only if the subsequent minimizer motion between $\hat{t}$ and the end of training is small.

Conceptually, we can think of a *drift rate* at each point in training, proportional to $\eta(\hat{t})h(\hat{t})$, which measures how fast the optimizer "forgets" earlier minimizers. The cumulative drift that erodes the utility of a gradient from fraction $\hat{t}$ is then the integral of this rate over the *remaining* training,

$$r(\hat{t}) \;\propto\; \int_{\hat{t}}^{1} \eta(s)h(s)\,ds.$$

Empirically, we find that $r(\hat{t})$ is well fit by a power law $\hat{t}^m$, as used in our predictive form for the TREC (Eq. (3)):

$$\hat{\mathcal{L}}_{\mathrm{re}}(\hat{t}) = 1 - c(\hat{t})^p\,\hat{t}^m,$$

where $c(\hat{t})$ is the AdamW EMA coefficient and $m$ captures the cumulative effect of curvature- and LR-driven drift along the training-fraction clock.

## J.7 EMPIRICAL EVIDENCE FOR VARIATION IN $m$

Interpreting the $\hat{t}^m$ term as a cumulative drift means that larger $m$ values (higher drift) correspond to gradients losing effectiveness more quickly; higher drift affects the TRECs by increasing the span of "forgotten" data—i.e., the data with baseline higher TREC loss (regardless of $c(\hat{t})$ coefficient).

From our analysis, $m$ should change under interventions that modify either the curvature/learning-rate product $\eta(\hat{t})h(\hat{t})$ or the optimizer's effective memory. Consistent with this prediction, we observe systematic variation in $m$ across multiple settings presented earlier in the paper:

**Lower EMA timescale.**  Shortening the AdamW EMA timescale $\tau$ decreases the optimizer's effective memory, and therefore increases the rate-of-change of parameters. This should lead to faster reductions in gradient effectiveness, with an effect on TRECs independent of the separate, analytical effect of a smaller $\tau$ on the EMA coefficients themselves. Indeed, we observe higher optimal $m$ values for lower timescales in our experiments (e.g., Fig. 6). Moreover, our $m^*$ power law fitted at 111M scale predicts well across model sizes: the $\tau$-governed drift is scale-invariant.

**Higher tokens-per-parameter (TPP).**  We also see a small but consistent increase in optimal $m$ when increasing the TPP ratio (also shown in Fig. 6). Note the normalized LR schedule $\eta(\hat{t})$ does not change with TPP. We therefore hypothesize that the degree of overtraining (as defined by TPP) slightly affects the pace of change in curvature (as a function of training fraction), which drives the pace that earlier gradients become ineffective. Yet it remains notable that these changes in $m$ are solely TPP-dependent and do not depend on the absolute number of training steps (which scales with model size for a given TPP).

**Fewer MoE experts.**  In sparse Mixture-of-Experts (MoE) models, *reducing* the number of experts *increases* the number of tokens processed by each expert's parameters, raising the expert's effective TPP. The impact on $m$ closely matches the shift observed when directly changing TPP in dense models, suggesting that MoE layers inherit the same drift scaling in TPP as non-expert dense layers.

**CPT vs. from-scratch training.**  Recall Fig. 7 comparing two 3.9B models trained with identical configurations—same data, batch size, weight decay, and an identical learning-rate schedule $\eta(t)$—but different starts: one from random initialization and one from a near-optimal checkpoint (i.e., this variant undergoes continual pre-training, CPT). Although $\eta(\hat{t})$ and EMA coefficients $c(\hat{t})$ were identical in both runs, the two trajectories occurred at different points in training fraction $\hat{t}$ and thus likely under different curvature statistics $h(\hat{t})$. Beyond curvature variation, changes in the extent of feature learning and evolving representations in earlier layers may also play a role. In a deep network, early updates are large compared to weight magnitudes (despite preconditioning) (Kosson et al., 2024), and thus optimization steps can produce disproportionately large shifts in the parameters, and thus the optimal batch minimizers. Empirically, the CPT model, starting in a region already well aligned with the final parameters, exhibited little minimizer drift: its $\hat{t}^m$ term remains close to 1, producing a TREC prediction closely following the EMA coefficients. In contrast, the scratch model traversed a much larger region of the loss landscape, with substantial drift that reduced alignment to early updates; its $\hat{t}^m$ term rises gradually from 0 to 1, with a corresponding TREC valley only aligning to the EMA coefficients near the very end of training.

**WSD vs. Cyclic LR schedules.**  In Appendix F.2, we compared the *WSD* and the *Cyclic* schedules. Training runs with the two schedules completely align in step-wise LR, weight decay, and batch size—in the final 20% of training. In a way, this is a similar test to the CPT vs. from-scratch comparison above: two similar training runs, but initializing from different start points. In this case, however, the two models have undergone a similar total amount of training during their initial phases, and consequently the initial gradients in the final phase have similar alignment with the final model parameters. Consequently, the final period spans the same range of training fraction, and their TRECs align very well over this final period. For the same peak LR, the *WSD* schedule does appear to drop slightly lower at around, e.g., $\hat{t} = 0.8$, which may reflect a more mature model at that stage (perhaps because this schedule, unlike *Cyclic*, does not have the periods where LR returns to zero).

**Different $m^*$ for different learning rate schedules.**  While the minor differences between *WSD* and *Cyclic* are unlikely to be consequential in terms of data placement strategy, differences in the final stages of training have a greater impact.

As we noted in Appendix G.4, optimal $m$ tends to change when the learning rate decay pattern changes (schedule-specific fits yield the best predictive performance). We also separately observed that *Constant* learning rate schedules produce the largest optimal $m$ values. This is consistent with

our drift interpretation: without decay, $\eta(\hat{t})$ remains high even late in training, inducing substantial movement in $\theta_T$ and its associated local minimizer. Earlier gradients are therefore misaligned more quickly, requiring a larger $m$ to match the observed TRECs.

Across these cases, the observed shifts in $m$ are consistent with changes in the effective $\eta(\hat{t})h(\hat{t})$ product. Together, these results suggest that a functional form may exist for local minimizer drift that not only normalizes across scale (as does the training fraction term), but also across LR schedules. That is, the functional form for this drift could directly incorporate the LR schedule itself.

