# OpenReview forum: "Predicting Training Re-evaluation Curves Enables Effective Data Curriculums for LLMs"
_ICLR.cc/2026/Conference — ICLR 2026 Poster_

### Official Review · Reviewer_YJr3 · 2025-10-21

**Soundness:** 3
**Presentation:** 3
**Contribution:** 3
**Rating:** 6
**Confidence:** 4

**Summary:**

This paper introduces the Training Re-evaluation Curve (TREC), a novel diagnostic tool for designing effective data curriculums for Large Language Models (LLMs). The TREC measures the retrospective loss of the final trained model on individual training batches, plotted as a function of when those batches were seen during training. The core hypothesis is that placing high-quality (HQ) data at the minimum of this curve—the "TREC valley"—maximizes its positive impact on the model's final performance.

The central contribution is a framework for predicting the TREC in advance, obviating the need for a costly initial training run. This prediction is grounded in the dynamics of the AdamW optimizer, modeling its parameters as an Exponential Moving Average (EMA) of weight updates. The authors extend this by incorporating a "training-fraction" term to phenomenologically account for "minimizer drift," where the loss landscape shifts during training, reducing the effectiveness of earlier updates. The paper provides extensive empirical validation for models ranging from 111M to 3.9B parameters, showing that the predictive model accurately captures TREC shapes and that TREC-guided data placement outperforms baseline strategies. Finally, the framework is applied to explain and critique existing training recipes and to improve performance in a large-scale continual pre-training setting.

**Strengths:**

1.  The TREC itself is a simple yet useful concept, and the ability to predict it proactively is a contribution.

2.  The claims are supported by a large-scale study of over many TRECs across a wide range of model sizes and training configurations.

3.  The framework successfully explains previously ambiguous empirical results from the community. For instance, it provides a clear rationale for why the late-stage annealing in Llama-3 405B failed to yield gains (the placement occurred on a rising part of the TREC) and why the mid-training interventions in models like OLMo-2 were effective.

**Weaknesses:**

1.  A significant scope limitation, explicitly noted by the authors (Appendix D.2, Figure 14), is that the framework can only identify the optimal data placement within a given learning rate schedule.

2.  How to explain the schedule with the highest learning rate (η=0.015) produced the deepest TREC valley, which according to the framework's core premise, should have yielded the best results. Instead, it produced the worst validation performance. This directly contradicts the assumption that minimizing re-evaluation loss on training data (what TREC measures) always translates to improved generalization.

**Questions:**

see weakness

---

> ### Author Response · Authors · 2025-11-20
> **Response to Reviewer YJr3 (1/1)**
>
> Thank you very much for your valuable feedback, and for emphasizing both the usefulness of TRECs and the breadth of our empirical study. We address your two main concerns below and will revise the paper accordingly.
>
> **1. Scope: within-configuration vs. across-configurations:** You are absolutely right to highlight the limitation around cross-schedule comparison, and your comment helped us realize that we need to be more precise in our terminology. In the current draft, we say that our framework "does not generalize across learning-rate schedules," but what we actually use in all experiments is a fixed optimizer configuration:
> - fixed LR shape η(t), peak LR η_max, weight decay λ, batch size B, and thus a fixed AdamW timescale τ.
>
> *Within* such a fixed configuration, our (validated) empirical claim is:
> - For a fixed optimizer configuration (η(t), η_max, λ, B, τ), placing HQ data at the TREC minimum yields the best performance among all placement positions *using that configuration*.
>
> This is what we validate across all placement experiments: at 610M scale (Fig. 1-2, 11-13) and in the 3.9B CPT experiments (Fig. 10), **including the η = 0.015 case, where placement at its own TREC minimum is still best**. What does *not* hold (and what Fig. 14 and the CPT runs with varying η make clear) is the stronger statement:
> - The configuration with the globally lowest *absolute* TREC valley (across different η_max / λ / B choices) will also yield the best validation loss.
>
> We agree this stronger claim is not supported by our results and we will clarify in the revision that we do not intend to make it. Specifically, we will:
> - Replace vague "within a schedule" phrasing with explicit "within a fixed optimizer configuration (η(t), η_max, λ, B, τ)," both in the Introduction and in Sec. 2/5;
> - Move a short version of Key Takeaway 5 from Appendix D.2 into the main text to explicitly say that cross-configuration selection is an open problem; and
> - Make it clearer that varying η in CPT is exactly an example of this across-configuration limitation, not a counterexample to the within-configuration result.
>
> **2. Why the η = 0.015 run has the deepest valley but worst validation**: Your question about η = 0.015 hits a genuinely interesting phenomenon, one that we have been giving much thought to recently.
>
> The TREC, by construction, measures how well the final model retains each training batch, i.e., a kind of "training-data retention" signal. Validation performance, however, depends on *how* that retention is achieved. Recent work on grokking and memorization (Powers et al., 2022; Morris et al., 2025) shows that models can reduce loss on seen data via:
> - **Memorization-dominated behaviour**: the model fits the specific examples it has seen, but generalizes poorly;
> - **Grokking/generalization**: the model begins to capture underlying structure and validation performance improves, sometimes only after extended overfitting on the training set.
>
> Both mechanisms can produce similarly-low re-evaluation loss, but obtain different validation performance. As we noted to Reviewer MKNN, we speculate that the pattern in Fig. 14 reflects exactly this phenomenon: placements in later, more grokking-dominated phases of learning tend to have *disproportionately better validation performance*. In the CPT studies, we also hypothesize that larger LRs move the model away from the grokking-dominated phase (reached after very long pre-training), and back into a more memorization-heavy regime.  Consequently, while the η = 0.015 configuration yields the deepest TREC valley, it generalizes worse to validation data.
>
> We agree this distinction between:
> - "Where should I place data within the training run I'm using?" versus
> - "Which optimizer configuration should I use in the first place?"
>
> needs to be made much more explicit. In the revision we will:
> - Explicitly call out the CPT case (including η = 0.015) as an example of a generalization gap across optimizer configurations;
> - Frame it as an interesting direction for future work: augmenting TRECs with additional signals (e.g., width of the valley, train-validation gap, curvature/flatness proxies) to distinguish memorization-heavy vs. genuinely generalizing valleys; and
> - Emphasize in the Limitations section that jointly choosing both placement and optimizer configuration is an open goal that we view as an important next step, not as something solved by the current paper.
>
> We really appreciate that your comments pushed us to sharpen this "within-configuration vs. across-configurations" distinction and to interpret the η = 0.015 result more clearly as a generalization story rather than a failure of the TREC idea itself.

---

### Official Review · Reviewer_79pF · 2025-10-28

**Soundness:** 3
**Presentation:** 2
**Contribution:** 2
**Rating:** 4
**Confidence:** 4

**Summary:**

This paper introduces the Training Re-Evaluation Curve (TREC) — a diagnostic that evaluates how well a trained model retains training batches as a function of when they appeared during training. TREC measures retrospective loss on training data using the final model weights and reveals that models tend to retain information from certain phases of training more effectively.
The authors show that placing high-quality data at the low points of the TREC (where retrospective loss is smallest) improves performance, contradicting the common practice of placing such data at the end of training.
They further demonstrate that TRECs can be predicted in advance from AdamW’s exponential moving average (EMA) timescale, enabling proactive curriculum design without retraining.
The paper supports its claims with both theoretical derivations and large-scale empirical results (from 111M to 3.9B parameters). It applies TREC analysis to settings such as sparse mixture-of-experts (MoE) models and continual pre-training (CPT), showing predictive consistency and practical gains.

**Strengths:**

- The paper provides a great amount of experiments on models from 111M to 3.9B with several hundred billions of tokens.
- The theoretical framing is well-grounded, extending prior work on AdamW dynamics and EMA analysis And the predictive model of TRECs is supported by strong correlations.

**Weaknesses:**

Since the experimental setup's setting defines HQ(high quality data) as a data with close distribution to the evaluation target ( e.g. code data vs general data and evaluating on code data, not general HQ data). Intuitively, place these data near the end of training will improve the performance as they are less likely to be forgotten by LLM, which is similar to annealing, and the results of the paper also show that place the data at the end of learning yields better results in linear decay cases.
Also for step scheduler, place the data at the end of learning (the last part before the lr step decay, as the remaining part may not change much on the parameters due to very low lr) yields better results. And TREC can be interpreted as a reflect how model remember/forget the data.

**Questions:**

1. The setting for each experiments may need further clarification or better organized. The pretraining loss has a sudden drop at 70% (i.e. figure 1 most left sub-figure), does that correspond to the experiment with Jump-to-1% learning rate scheduler?
2. Is there a specific kind of data (i.e. domains or other patterns) tend to have lower TREC? Or it is just related to the order across the whole training?

---

> ### Author Response · Authors · 2025-11-20
> **Response to Reviewer 79pF (1/1)**
>
> Thank you very much for your insightful comments, and for highlighting the extensive experiments, the well-grounded theory, and the strength of the predictions.
>
> **1. Clarifying the core empirical message**:  Yes, intuitively HQ data should not be forgotten and might perform well if placed late, similar to "data annealing." Our intent is not to argue that "late placement is bad," but to make a sharper statement:
> - **Within a given schedule, the optimal placement corresponds to the TREC valley, not necessarily the literal end of training, nor the point just before the LR becomes very small.**
>
> Concretely:
> - For the **decay-to-zero** schedule with 10% HQ data, the best *decile* is indeed the last one. However, Fig. 11 shows that the TREC bends back up as LR approaches zero; if we allowed *arbitrary* placement rather than restricting to fixed deciles, the optimal window would lie slightly *before* the very end of training.
>
> - For the **Step** schedule, the point you describe as "the end of learning" (just before the LR drop) is precisely where the TREC valley occurs. Importantly, the subsequent low-LR region still **changes the parameters**: the vanilla training loss drops substantially (from 2.9 to 2.7). Looking only at this curve, many practitioners might previously have argued for annealing **during** this rapidly-improving phase, but the TREC (and our placement experiment) shows this intuition is incorrect.
>
> - In the **3.9B CPT** experiment (Fig. 10), all runs use a linear decay-to-zero schedule. Nonetheless, placing HQ data *midway* through CPT consistently **outperforms placing it at the very end**, across all tested LRs.
>
> Overall, the TREC provides a principled way to identify where placement matters, rather than relying on intuition about when the LR is too small. We will revise Sec. 2 to emphasize this "late but not necessarily last" message, and to distinguish between "end of training," "end of learning," and the TREC valley.
>
> **2. Clarifying experimental settings and Fig. 1**: You are absolutely correct that the experimental setup and figure organization can be clearer:
> - Yes, the sudden drop around 70% in the left panel of Fig. 1 corresponds to the Jump-to-1% LR schedule.
>
> As we noted to Reviewer wgGG, we will replace informal names like “10×” with explicit descriptions such as “linear decay to 0.1 × peak LR”, and “Step decay” with “single step drop at 70% of training from ηmax to ηmin", etc. We will systematically clarify the corresponding figure captions. We will also add a small table in Section 2 summarizing each LR schedule (η(t), plot reference, associated figures, and key takeaway) to make it easier to track which result corresponds to which LR schedule.
>
> **3. Do some kinds of data have lower TREC, or is it purely order?** This is a great question, and we agree it deserves to be highlighted better. In our current experiments:
> - There is substantial variation in raw TREC loss *across batches*, reflecting **inherent batch difficulty**. For example, in Fig. 1, the Code Blend has lower loss than the General Blend; batches that happen to contain more code tokens will typically have lower TREC loss than batches that are mostly general text.
> - To specifically focus on order, we therefore **average across batches** when plotting TRECs. This deliberately "homogenizes" the data so that the *shape* of the curve primarily reflects retention as a function of training position.
>
> For domain-specific TRECs, our working hypothesis is that while some domains have systematically lower *absolute* loss, their **normalized TRECs would look very similar**: that is, TREC *shape* is driven mainly by the optimization dynamics (LR schedule, τ, etc.), rather than the particular mix of data. This is why a TREC measured on the aggregate blend still successfully guides where to place code or math data.
>
> Your question goes one step further: are there domains that are unusually well retained on re-evaluation compared to their unseen-validation performance (i.e., that receive an extra "retention boost")? We did not explore this here, but agree it is a very promising direction. In the revision we will:
> - explicitly state that our analysis averages over domain/batch variability in order to isolate order effects, and
> - add to the Limitations / Future Work section that studying domain-specific TRECs and cross-domain comparisons (e.g., code vs. math vs. web) is an important next step.
>
> Thanks again for your thoughtful comments; they directly shaped how we will clarify the setup and better communicate the relationship between TREC, annealing-style heuristics, and optimal placement.

---

### Official Review · Reviewer_MKNN · 2025-10-30

**Soundness:** 3
**Presentation:** 4
**Contribution:** 3
**Rating:** 6
**Confidence:** 4

**Summary:**

This paper introduces the Training Re-evaluation Curve (TREC), a novel diagnostic tool for understanding data retention in Large Language Model (LLM) training. The TREC is computed retrospectively by evaluating the final trained model on the sequence of training batches it encountered. The authors hypothesize that the optimal placement for high-quality (HQ) data in a curriculum is at the minimum of this curve (the "TREC valley"), a point that often precedes the end of training.

The central challenge is that TREC is a retrospective measure. The paper's main contribution is demonstrating that the TREC's shape is predictable in advance. The authors show that TREC shape is governed by the AdamW optimizer's implicit EMA timescale and a "training-fraction" component. They develop a predictive model for TREC and show that its parameters can be fitted on small models (e.g., 111M) and generalized to larger ones (up to 3.3B).

The authors validate this framework by showing that:

1. Directly placing HQ data at the TREC minimum in 610M model runs yields the best validation performance.

2. The framework successfully explains findings in prior work, such as the (ineffective) late-stage annealing in Llama 3 405B and the (effective) data phase timing in OLMo-2.

3. It provides a compelling explanation for memorization in Mixture-of-Experts (MoE) models, linking it to a reduced "effective TPP" for each expert.

4. TREC-guided placement improves performance in a 3.9B model continual pre-training (CPT) task.

**Strengths:**

1. **Originality and Significance:** The paper's primary strength is its originality and practical significance. It defines a novel diagnostic (TREC) and, crucially, provides a complete, principled framework to *predict* it in advance. This transforms curriculum design from a heuristic, high-cost ablation study into a predictable, low-cost design choice. This is a significant step forward for the field.

2. **High-Quality, Thorough Experimentation:** The empirical work is extensive and of high quality. The authors support their claims with a large study of models (111M to 3.9B) , diverse experimental settings (pre-training, CPT, MoE) , and a large-scale analysis (over 600 TRECs). The "data placement sweep" (Fig. 2) is a direct, clear, and compelling validation of the core hypothesis.

3. **Presentation:** As noted above, the paper is exceptionally clear. The logical flow (Here's a tool -> Here's the problem -> Here's the solution -> Here's why it works -> Here's how to use it) is perfectly executed. The figures are exemplary.

4. **Actionable Insights:** The framework is not just theoretical; it provides immediate, actionable insights. The explanations for MoE memorization (linked to effective TPP) and the failure of Llama 3's late-stage annealing (linked to vanishing EMA coefficients) are powerful demonstrations of the framework's explanatory power. This gives confidence that the model is capturing a true, underlying dynamic of the training process.

**Weaknesses:**

1. **Limited Scale of Core Validation:** The primary weakness is that the main *predictive loop* (fit on small model, predict for large model, train large model based on prediction) is not fully closed at the largest scales.

    - The core *validation* of the hypothesis (the data placement sweep in Fig. 2) was done on a 610M model.

    - The *prediction* framework ($m^\*$) was fit on 111M and *evaluated* on runs up to 3.3B.

    - The 3.9B CPT experiment is the closest to a full-loop validation. However, it's a CPT setting, not pre-training from scratch. Furthermore, it's not explicitly stated whether the "Halfway" point was *proactively chosen* by the 111M-derived predictor, or if it was just a reasonable choice *inspired* by the general shape of TRECs.

    - To be fully convincing, a stronger validation would be to use the 111M-derived $m^\*$ predictor to find the optimal placement for a 7B+ *pre-training* run and show it outperforms the standard heuristic.

2. **Cross-Schedule Generalization Limitation:** The paper is commendably honest about this limitation, but it is a significant one. The framework can reliably find the best data placement *within* a given LR schedule. However, it *cannot* be used to compare *across* schedules.

**Questions:**

1. **On the 3.9B CPT Experiment:** In the 3.9B CPT experiment (Sec. 5.3), was the "Halfway" placement point proactively calculated using the power-law predictor (Eq. 4) that was fit on the 111M model? Or was this point chosen based on a retrospective analysis of a 3.9B run, or simply as an intuitive "non-end" baseline? Clarifying this is critical for understanding the full practical utility of the predictive framework at scale.

2. **On Cross-Schedule Comparison:** The finding that absolute TREC loss does not predict performance *across* LR schedules (Fig. 14, Key takeaway 5) is an important limitation. Do the authors have any intuition on what additional signal might be required to make such a cross-schedule prediction? For example, does the absolute value of the *training loss* (the orange curve) at the TREC valley, or the *width* of the valley, play a role in this disconnect?

3. **On Predictor Robustness:** The $m^\*$ power-law was fit using a Linear decay schedule, and the paper notes a Cosine-specific fit is more accurate. How sensitive is the framework to the LR schedule shape? If a practitioner wishes to use a more complex schedule (e.g., WSD, or a custom multi-stage schedule), must they re-fit the entire power-law (which requires many small-scale runs)? Or is the provided Linear-fit predictor "good enough" for most practical purposes to identify the approximate valley?

---

> ### Author Response · Authors · 2025-11-20
> **Response to Reviewer MKNN (1/1)**
>
> Thank you so much for your detailed and valuable review, and for your very kind words regarding the paper's originality, practical significance, extensive/high-quality experimentation, and "exceptionally clear" presentation. We address your main questions below.
>
> **1. On the "Halfway" placement in the 3.9B CPT experiment**: Thank you for highlighting this; we agree it should be clarified. The "Halfway" placement was *not* chosen directly from the 111M-derived predictor. Before running the data-placement experiments, we first computed the empirical TREC for the vanilla CPT run (orange line in Fig. 7, right), and it was this measured TREC (along with checkpoint availability at that point) that informed our placement decision. We will revise the paper to state this explicitly. Importantly, the empirical and predicted TRECs for the 3.9B run match well (Fig. 7), so had we used the predictor directly, it would have recommended a similar placement.
>
> More broadly, the paper introduces TRECs and studies three scientific questions:
> 1. whether TRECs effectively guide placement,
> 2. how TREC shape varies with hyperparameters and scale, and
> 3. whether TRECs can be predicted in advance.
>
> The 610M and 3.9B placement-sweeps empirically validate 1., and results in Section 3 and 4 validate 2. and 3. As noted briefly in lines 233-234, practitioners also have a second predictive option: re-evaluating a smaller model with matching τ, TPP, and LR schedule, which our experiments show transfers well across scale. The analytical power-law predictor and the small-model re-evaluation thus provide two complementary ways to predict TRECs in practice.
>
> We agree that a single 7B+ from-scratch run demonstrating "prediction + prediction-guided placement" would represent a full-loop validation. Unfortunately this is outside our compute budget and rebuttal timeline. However, the core takeaway—that TREC-guided placement consistently improves upon heuristic end-placement—holds regardless of which predictive path is used.
>
> **2. On cross-schedule comparison**: We appreciate this question; it points directly to a deeper explanation that we have recently been considering. Indeed, while TREC loss correlates well with validation loss *within* a schedule, two schedules A and B may attain the same TREC loss at certain positions, yet A may *generalize* better than B. A possible explanation is that A and B can achieve similar TREC loss via *qualitatively different mechanisms*. Recent grokking studies (Powers et al., 2022; Morris et al., 2025) show that models initially reduce training loss through **memorization**, and only later transition into **grokking**, where they learn structure that generalizes. Both mechanisms can yield equally low TREC loss, but only grokking produces strong *validation* performance.
>
> This provides a natural interpretation of why placement at position 10 of the 10× schedule generalizes better than early decay-to-zero placement, even though both achieve similar TREC losses: the 10× schedule evidently leverages more grokking and less memorization. Likewise, in the CPT LR sweeps, high LR appears to push the model back into a memorization regime, while low LR preserves grokked structure and yields better validation loss.
>
> These observations connect to classical factors known to influence generalization (such as curvature, SGD noise, and basin geometry), which may vary systematically across LR schedules. If we can identify signals that distinguish memorization-driven vs. grokking-driven TREC improvements (e.g., the width of the TREC valley, or deviations between the TREC and the standard training-loss curve), we could potentially correct for these generalization differences and enable the cross-schedule predictor you suggest. We will expand the Limitations section to highlight this as a promising direction for future work.
>
> **3. On predictor robustness and schedule-specific fits**:
> - As noted in lines 233-234, one can obtain a high-fidelity TREC by re-evaluating a small model with matching τ, TPP, and LR schedule. This requires only one small-scale run and directly provides placement guidance for the larger model. We will highlight this option more explicitly in the paper.
> - Regarding whether the Linear-decay fit is "good enough", it depends on practitioner goals. Researchers routinely rely on high-level guidance derived from related settings (e.g., Chinchilla's 20-TPP rule) even without re-optimizing for their exact distribution. We believe the Linear-fit predictor serves a similar role: a strong out-of-the-box guide for most users, and far more informative than the default end-of-training heuristic. For large-scale production training, hyperscalers could indeed re-fit schedule-specific TREC predictors at small scale (analogous to how they re-fit scaling laws). We will clarify this tradeoff in the paper.
>
> Thank you for helping us think more deeply about this and all the other points. These are excellent refinements!

---

### Official Review · Reviewer_wgGG · 2025-11-02

**Soundness:** 3
**Presentation:** 3
**Contribution:** 2
**Rating:** 8
**Confidence:** 4

**Summary:**

This paper introduces the Training Re-Evaluation Curve (TREC), a diagnostic tool that measures a trained LLM's retention of training batches as a function of their position, revealing optimal placements for high-quality data that outperform end-of-training heuristics. By modeling TRECs analytically via AdamW's EMA coefficients adjusted for training fraction drift, the authors enable proactive curriculum design, with predictions generalizing across scales from 111M to 3.9B parameters and 20-1280 TPP. Extensive experiments explain suboptimal strategies in prior works like Llama-3 and OLMo-2, while demonstrating TREC-guided placement improves continual pre-training in validation loss.

Overall, I really like this paper. Meanwhile, as a practicer in this field, I have many questions and want to discuss with authors. I am looking forward to replies from the authors.

**Strengths:**

1. The paper addresses a critical challenge in LLM pre-training from an innovative and unanticipated perspective, demonstrating strong novelty that sets it apart in the field.
2. The authors conduct a comprehensive suite of rigorous experiments, involving substantial computational resources, which will provide valuable insights to researchers in the LLM pre-training community.
3. I particularly appreciate the paper's clear and logical structure: it begins with well-defined hypotheses, builds through theoretical derivations, validates them empirically, and systematically distills a series of key takeaways.

**Weaknesses:**

1. **Difficulty Understanding Loss Variations in TREC**: What I find challenging to comprehend is the section defining TREC. For a given data distribution, different training batches should be nearly identically distributed (i.i.d.). Why, then, does the fully trained model exhibit such substantial loss differences across these batches—particularly in Figure 1 (left) and Figure 2 (left)? In my view, the losses should approximate a flat line parallel to the x-axis, with values closely aligning with the final train loss. This discrepancy warrants further clarification.

2. **Scope of TREC's Applicability**: It appears that this principle applies only to validation sets identical to the training distribution, since TREC is computed on training batches. Is my interpretation correct? If so, does this imply that the optimization ultimately targets only validation sets matching the training distribution, rather than arbitrary domain-specific tasks (e.g., code)? Your statement on line 097—"we hypothesize the TREC remains predictive even when data distributions vary"—seems to address this concern. However, this hypothesis comes with no more explanation. Is this hypothesis overly ambitious, given the potential risks of distribution shift?

3. **Inconsistent and Unclear Naming of LR Schedules**: I feel that your selection and nomenclature for LR schedules lack rigor and clarity, manifesting in several ways:
   a. Terms like "10x" and "step decay" are non-standard for LR schedules, and I have rarely encountered similar naming conventions in other literature. While I eventually deciphered "10x" via the appendix, this remains confusing and detracts from accessibility.

   b. "Step-decay learning rate schedule" is also uncommon; I am still unclear on its precise form. Does it entail a sudden drop in LR (e.g., at 70% of training) to 10% of its prior value? You reference it "as used in DeepSeek LLM," but DeepSeek employs a "multi-step cosine scheduler," which introduces ambiguity.

   c. I strongly recommend adopting more precise terminology for your experimental LR schedules, ideally with explicit mathematical functions describing LR evolution at each step.

4. **Summary of Predicted TREC Positions Across Schedules**: Allow me to summarize the optimal high-quality data placement positions predicted by your TREC across different LR schedules:
   a. Linear decay (commonly used): End of training (Figure 2 right).
   b. Decay to zero (commonly used): End of training (Figure 11 left).
   c. Decay to 0.1x LR (your "10x," commonly used): Not explicitly described, but I infer it aligns with the end, given its similarity to D2Z.
   d. Step drop (uncommonly used): Pre-decay position (Figure 2 left), with the final val loss showing minimal gap compared to end placement.
   **Thus, can I conclude that, for the vast majority of common LR schedules, TREC predicts the end of training as the optimal placement?** This suggests some degree of over-claiming on your part, while also raising concerns about TREC's practical utility in experiments. I acknowledge that TREC offers mechanistic insights, but its impact on guiding real-world practice seems limited.

5. **Alignment in Figure 9 and Practical Implications**: Following the previous point, in Figure 9, your TREC predictions align with prior findings, yet these still involve placing high-quality data in the final stages. Notably, in Figure 9 left, your prediction actually points to roughly the 0.5–0.9 fraction for inserting that 40% high-quality data, rather than purely the end (0.6–1.0 fraction). I am skeptical: under a common LR schedule, would shifting high-quality data away from the very end—per TREC guidance—truly yield superior performance **in practice**? This merits empirical validation beyond the current analysis.

**Questions:**

All the questions are recorded in weaknesses sections.

I greatly enjoyed this paper. If the authors can thoughtfully address and resolve my concerns, I would be delighted to raise my score to a 10. Conversely, given the fundamental nature of my questions—particularly the first one—if they remain inadequately resolved, I may lower my score to 6 or even below. I look forward to further discussion with the authors during the rebuttal phase.

---

> ### Author Response · Authors · 2025-11-20
> **Response to Reviewer wgGG (1/2)**
>
> Thank you very much for your thoughtful and encouraging review, and for engaging so deeply with the paper. We are especially grateful for your comments on the paper's strong novelty, logical structure, rigorous experiments, and valuable insights. Below we address your main concerns and outline how we will clarify the paper.
>
> **Clarifying loss variations in TRECs**: Yes, clarifying this will definitely strengthen the paper; thank you for highlighting it. We propose to add the following discussion after Definition 1:
>
> > It is worth clarifying why re-evaluation loss may depend on order, even if the training batches themselves are drawn i.i.d. Fundamentally, for any online optimization algorithm, each training batch will be seen at a different point in the training trajectory, with a different parameter state, and a different amount of parameter evolution *after* the batch is seen. Early batches are learned when the parameters are still far from their final state, and directional parameter updates derived from these batches (gradients) lose their effect as the parameters subsequently evolve to different regions in the loss landscape. The result is higher re-evaluation loss on earlier batches: the model performs these *forgotten* data similarly to how it performs on unseen validation samples. This is also closely related to phenomena such as *catastrophic forgetting* observed in continual learning: earlier examples can be forgotten, and the position in training matters *even if the input distribution is unchanged.* Furthermore, as far back as Graves (2013), it was observed that generated text from a neural language model tends to inordinately reflect the final training batches; from a TREC-perspective, this manifests as lower TREC losses on the final training samples. In Appendix J, we formalize these effects and show how TREC loss can be *analytically* predicted as a function of parameter evolution, under a simplified quadratic model.
>
> This directly addresses why TRECs are not expected to be flat even under i.i.d. sampling and ties the intuition to the theory in Appendix J.
>
> **TREC's applicability across data distribution**: You are right that our statement "we hypothesize the TREC remains predictive even when data distributions vary" needs elaboration. Our intended claim is that **TREC shape is driven by optimization dynamics, not by the distribution used to compute it.** More concretely, our *experimental* hypothesis is:
> - while the canonical TREC is defined under homogeneous sampling from a base distribution,
> - the relative ranking of positions induced by that TREC (which segments are better retained vs. more forgotten) remains predictive when we inject *different* data at those positions.
>
> This is exactly how we use TREC in our heterogeneous setting. We:
> - compute the TREC on one kind of data (e.g., the *aggregate blend*), and
> - use it to effectively guide where we place another kind of data (e.g., the *code blend*).
>
> Our results confirm that code placements are indeed most effective at low points in *aggregate blend* loss (Fig. 1 left, Fig. 2 right, Figs. 11-12), while removing *general blend* data at those positions also most impairs *general blend* loss (Fig. 13). In our CPT experiments, we also show that re-evaluation loss on the general blend correlates perfectly with re-evaluation loss on placed math data (Fig. 24, left).
>
> We will revise the paper to provide the above elaboration and, after presenting the experimental results, explain how they validate the "remains predictive when distributions vary" hypothesis, while noting that our current evidence is limited to related blends (code vs. general text) rather than arbitrary domain shifts.
>
> **Inconsistent and Unclear Naming of LR Schedules**: We appreciate this feedback and agree that our LR schedule naming should be more standard and explicit. We will replace informal names like “10×” with explicit descriptions such as “linear decay to 0.1 × peak LR”, and “Step decay” with “single step drop at 70% of training from ηmax to ηmin", etc. We will systematically clarify the corresponding figure captions. We will also add a small table in Section 2 summarizing each LR schedule (functional form η(t), plot reference, associated figures, and key TREC takeaway (e.g., "position 7 is the TREC minimum for this schedule")).
>
> Regarding DeepSeek: we referenced it as an example of large-scale training that uses discrete LR drops. We will clarify that our "Step" schedule is intentionally much simpler than the "multi-step cosine" scheduler used in DeepSeek (we were loosely inspired by the idea of a late LR drop). Notably, Figure 1(a) in Bi et al. (2024) shows a sharp drop in training loss at 80%, similar qualitatively to the drop we observe in our own Fig. 1 (left); we will make this connection more precise and avoid any implication that the schedules are identical.

---

> > ### Author Response · Authors · 2025-11-20
> > **Response to Reviewer wgGG (2/2)**
> >
> > **TREC-guided positions versus default end-placement**: Your point is well taken, and we agree this is an important nuance to clarify:
> >
> > First, considering the Sec. 2 tests:
> > - When training is divided into ten disjoint 10% segments, then under linear decay-to-zero, the best segment is indeed the last decile.
> > - However, as Fig. 11 shows, the TREC minimum is not *exactly* at the very end: the curve bends back up as LR goes to zero. If we allowed arbitrary placement (not constrained to deciles), the optimal 10% window would lie slightly **before** the literal end of training.
> > - Moreover, at larger scales, it is rare to have a full 10% of pre-training tokens be "highest-quality" data (e.g., *trillions* of HQ tokens). For smaller HQ budgets, the TREC shape implies that sub-segments earlier within that last decile (or even earlier in training, depending on the schedule) are better than placing at the very end, where LR is negligible.
> >
> > We will revise Sec. 2 to explicitly highlight this "late but not necessarily last" message.
> >
> > Second, regarding practical implications and your question about shifting away from the very end:
> > 1. In our 3.9B CPT experiment (Fig. 10), CPT uses a linear decay-to-zero schedule, yet placing HQ data midway through CPT **systematically outperforms placing it at the end** across all tested learning rates. This shows that shifting HQ data away from the very end *can* yield superior performance in a large-scale setting.
> > 2. In our analysis of Llama-3 405B, the GSM8k/MATH annealing phase occurs when EMA coefficients are essentially zero. This is another real-world example where placing data at the very end **fails to provide gains**; the TREC is already back at baseline.
> >
> > We will add a forward pointer to these findings in Sec. 2, and will acknowledge that further empirical validation is warranted (as you noted). Nonetheless, these results provide strong motivation for other practitioners to likewise test TREC-informed placements, rather than follow the default strategy.
> >
> > Thanks again for your detailed feedback; addressing these issues will strengthen the paper significantly.

---

### Author Response · Authors · 2025-12-02
**Author Summary**

We sincerely appreciate the reviewers' thoughtful, constructive feedback and the (new) area chair's extra effort in shepherding this paper under the challenging circumstances of the OpenReview incident.

Our paper evaluates fully-trained LLMs on their original training data, measuring retention across training steps. We call the resulting measurement the *training re-evaluation curve* (TREC). A predictive model of this curve (derived from AdamW EMA dynamics plus a training-fraction term) identifies optimal positions for high-quality data: TREC-guided placements outperform default end-of-training placement in multiple settings (including 3.9B-scale CPT). Reviewers were consistently positive about the novelty and usefulness of TRECs, the breadth and rigor of the experiments, and the framework's ability to explain existing training recipes.

In response to reviewer feedback, we will make three main sets of revisions:
- **Clarity and presentation (raised by wgGG, 79pF)**: We will add an intuitive paragraph explaining why TRECs vary with order even under i.i.d. sampling, clarify how TRECs are used with heterogeneous/domain-specific data, standardize LR-schedule naming, and add a small table summarizing each schedule's formula, associated figures, and key takeaway.
- **"Late but not necessarily last" placement (raised by wgGG, 79pf)**: We will clarify that, for common decay-to-zero schedules, the unconstrained optimum lies slightly *before* the very end of training, and explicitly highlight the 3.9B CPT result where mid-CPT placement outperforms end placement, addressing concerns that TRECs merely rediscover "put HQ data at the end."
- **"Scope of the predictive framework" (raised by MKNN, YJr3)**: We will state explicitly that our validated claim is: within a fixed optimizer configuration (η(t), ηmax, λ, B), the TREC minimum gives the best placement. We will move the "Key Takeaway 5" limitation on cross-configuration comparisons into the main text and briefly discuss a grokking-vs-memorization view that helps explain why configurations with similarly low TREC loss can differ in validation performance.

We are grateful for the feedback and believe the planned revisions will substantially improve the clarity and impact of the final version.

---

### Meta-Review · Area_Chair_jVtu · 2026-01-07

**Summary:**

Overall, reviewers were largely positive about the paper’s novelty, technical soundness, and experimental breadth. The primary concerns centered on **clarity, scope, and positioning**, rather than correctness.

Several reviewers (wgGG, 79pF) initially found it unintuitive that re-evaluation loss varies significantly across training steps even under i.i.d. data, and requested clearer explanation of why TRECs are not flat. The rebuttal provides a coherent optimization-based explanation (forgetting, EMA dynamics, minimizer drift) and commits to clarifying this in the main text.

A recurring limitation raised by multiple reviewers (MKNN, YJr3) was **scope**: the framework is validated for identifying optimal data placement *within a fixed optimizer configuration*, but does not support comparisons across learning-rate schedules or optimizer settings. The authors acknowledge this explicitly, refine the claims accordingly, and move this limitation into the main text.

Some reviewers questioned whether TREC-guided placement meaningfully improves over standard “place HQ data at the end” heuristics under common LR schedules. The rebuttal clarifies the “late but not necessarily last” message and points to empirical evidence, including the 3.9B CPT experiment, where non-end placement yields consistent gains.

Concerns about learning-rate schedule naming and presentation were largely editorial and are addressed via planned revisions (standardized terminology, explicit formulas, and summary tables).

The lowest-scoring reviewer (79pF) did not raise fundamental technical objections; their concerns focused on interpretation, clarity, and experimental organization, all of which are substantially addressed in the rebuttal and revision plan.

**Reviewer Concerns:**

**Concerns largely addressed in the rebuttal**

- **Why TRECs are not flat under i.i.d. data** (wgGG–W1, 79pF–W1):
  Reviewers initially found it unintuitive that re-evaluation loss varies strongly across training steps under i.i.d. sampling. The rebuttal provides a clear optimization-based explanation (forgetting, EMA dynamics, minimizer drift) and commits to adding an intuitive discussion in the main text, addressing this concern.

- **Applicability of TRECs beyond the training distribution** (wgGG–W2):
  The reviewer questioned whether TREC-guided placement applies only when the evaluation target matches the training distribution, and whether the claim about robustness to distribution shifts was insufficiently justified. The rebuttal clarifies that TRECs are used to rank *training positions* rather than data types, and that this ranking empirically transfers to placing related domain-specific data (e.g., code, math). The authors acknowledge the current evidence is limited to related domains and commit to clarifying this scope.

- **Interpretation of “late but not necessarily last” placement** (wgGG–W4, 79pF–W1):
  The rebuttal clarifies that optimal placement corresponds to the TREC minimum, which may occur slightly before the literal end of training under common schedules. This is supported by placement sweeps and the 3.9B CPT experiment, addressing concerns that the method merely recovers standard end-of-training heuristics.

- **Learning-rate schedule naming and presentation clarity** (wgGG–W3, 79pF–W2):
  These concerns were primarily about clarity and organization. The authors commit to standardizing terminology, explicitly defining schedules, and adding summary tables to improve readability.

---

**Remaining limitations / partially outstanding concerns**

- **Limited cross-configuration guidance** (MKNN–W2, YJr3–W1):
  Reviewers noted that while TREC reliably identifies the best placement within a given training setup, it cannot be used to compare or choose among different optimizer or learning-rate configurations. The rebuttal explicitly clarifies and scopes this limitation, but it remains an inherent constraint of the framework.

- **Incomplete full-loop validation at largest scale** (MKNN–W1):
  While placement sweeps at 610M and CPT results at 3.9B support the core hypothesis, the rebuttal confirms that the largest-scale placement was informed by an empirical TREC rather than a purely small-model prediction. A fully proactive large-scale validation remains untested due to compute constraints.

- **Retention vs. generalization distinction** (YJr3–W2):
  The rebuttal offers a plausible interpretation (memorization vs. grokking) for why deeper TREC valleys do not always correspond to better validation performance across configurations, but this remains an open question framed as future work.

**Reviewer Scores:**

- **wgGG (8)**:
  Strongly positive throughout. Main concerns were clarification and scope, which are directly addressed in the rebuttal. Score would likely remain **8**.

- **MKNN (6)**:
  Generally positive, with concerns about cross-configuration scope and large-scale full-loop validation. The rebuttal clarifies and appropriately scopes these limitations. Score would likely remain around **6**.

- **YJr3 (6)**:
  Raised concerns about cross-configuration generalization and retention vs. validation performance. These are clarified and reframed in the rebuttal, though not fully resolved. Score would likely remain around **6**.

- **79pF (4)**:
  No fundamental technical objections; concerns focused on interpretation, experimental clarity, and practical impact. These are largely addressed in the rebuttal. Score could reasonably increase to **5–6**.

**Overall**, based on reviewer tone and rebuttal discussion, a reasonable post-discussion interpretation of the reviews is approximately **8, 6, 6, and 5**, corresponding to an **accept** profile.

---

### Decision · Program_Chairs · 2026-01-26

Accept (Poster)